# EVIDENTIAL CONSERVATIVE Q-LEARNING FOR DYNAMIC RECOMMENDATIONS

## ABSTRACT

Reinforcement learning (RL) has been leveraged in recommender systems (RS) to capture users' evolving preferences and continuously improve the quality of recommendations. In this paper, we propose a novel evidential conservative Q-learning framework (ECQL) that learns an effective and conservative recommendation policy by integrating evidence-based uncertainty and conservative learning. ECQL conducts evidence-aware explorations to discover items that are located beyond current observation but reflect users' long-term interests. Also, it provides an uncertainty-aware conservative view on policy evaluation to discourage deviating too much from users' current interests. Two central components of ECQL include a uniquely designed sequential state encoder and a novel conservative evidential-actor-critic (CEAC) module. The former generates the current state of the environment by aggregating historical information and a sliding window that contains the current user interactions as well as newly recommended items from RL exploration that may represent future interests. The latter performs an evidence-based rating prediction by maximizing the conservative evidential Q-value and leverages an uncertainty-aware ranking score to explore the item space for a more diverse and valuable recommendation. Experiments on multiple real-world dynamic datasets demonstrate the state-of-the-art performance of ECQL and its capability to capture users' long-term interests.

## 1 INTRODUCTION

Recommender systems (RS) have been widely used for providing personalized recommendations in diverse fields, such as media, entertainment, and e-commerce by effectively improving user experience (Su & Khoshgoftaar, 2009; Sun et al., 2014; Xie et al., 2018). Most existing RS methods model recommendation as a static process, and therefore they cannot consider users' evolving preferences. Some efforts have been devoted to capturing users' evolving preferences by shifting the latent user preference over time (Koren, 2009; Charlin et al., 2015; Gultekin & Paisley, 2014). Similarly, sequential recommendation methods (Kang & McAuley, 2018; Tang & Wang, 2018) attempt to incorporate users' dynamic behavior by leveraging previously interacted items. However, both the above-mentioned static and dynamic recommendation methods primarily focus on maximizing the immediate (*i.e.,* short-term) reward when making recommendations. As a result, they fail to take into account whether these recommended items will lead to long-term returns in the future, which is essential to maintaining a stable user base for the system in the long run.

Several recent works have adapted reinforcement learning (RL) in the RS context (Chen et al., 2019b; Zhao et al., 2017). RL has gained huge success in diverse fields, such as robotics (Kober et al., 2013) and games (Silver et al., 2017). The core idea of RL is to learn an optimal policy to maximize the total expected reward in the long run. RL methods consider a recommendation procedure as sequential interactions between users and RL agents to learn the optimal recommendation policies effectively. Although RL approaches show promising results in RS (Chen et al., 2019b; Zheng et al., 2018), they primarily rely on standard exploration strategies (*e.g.,* $\epsilon$-greedy), which are less effective in a large item space with sparse reward signals given the limited interactions for most users. Therefore, they may not be able to learn the optimal policy that captures effective user preferences and achieves the maximum expected reward over the long term.

Figure 1 further illustrates the limitation of existing RL methods using a standard $\epsilon$-greedy strategy for exploration. The existing RL agent primarily focuses on highly-rated items in early steps, as shown

in Figure 1a. Most of these items come from a narrower set of genres as shown in Figure 1b, where different genres are denoted as Adventure (A), Drama (D), Comedy (C), Thriller (T), and Others (O).

These only represent the user's temporary interest as in the later steps the positive count (See definition in Appendix E) of the same genre is decreased. Such a recommendation behavior leads to a lower cumulative reward (average rating of recommendation) in the later steps. As Table 1 shows, $\epsilon$-greedy mostly focuses on Drama movies based on the user's current preference. It only captures one novel genre (*i.e.,* Musical, bold in the table) that matches the user's long-term interest. It clearly indicates that more systematic exploration is essential to discover users' long-term and diverse interests to maximize future rewards.

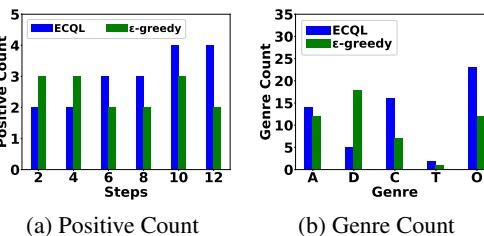

(a) Positive Count  (b) Genre Count

Figure 1: Different recommendation behavior between an existing RL model with $\epsilon$-greedy exploration strategy and our model ECQL with evidential uncertainty (*i.e.,* vacuity).

To address the above key challenges, we conduct novel evidential conservative Q-learning (ECQL) that utilizes a balanced exploitation (with high predicted ratings) and exploration (with evidential uncertainty) strategy for effective recommendations. We formulate an evidential RL framework that augments the maximum reward RL objective with evidential uncertainty to facilitate the exploration of unknown items. More importantly, the evidential uncertainty formulation substantially improves exploration and

Table 1: Examples of recommended movies

| Model | Important Items (Movies) | Movie Genre |
|---|---|---|
| ECQL | Sound of Music (1965) | **Musical** |
| | Casino (1995) | Drama |
| | Ben-Hur (1959) | **Action,Adventure** |
| | The Bug's Life (1998) | **Animation,Comedy** |
| | Babe (1995) | **Children's,Comedy** |
| $\epsilon$-greedy | Pocahontas (1995) | **Musical** |
| | Wizard of Oz (1939) | Drama |
| | Christmas Story (1983) | Drama |
| | Erin Brockovich (2000) | Drama |
| | Restoration (1995) | Drama |

robustness by acquiring diverse behaviors that are indicative of a user's long-term interest. As shown in Figure 1, ECQL devotes a strong focus on more diverse genres (*i.e.,* 'others' in the figure), and many of these capture the long-term interest from the user as verified by the detailed recommendation list in Table 1. Additionally, we encourage the model to explore items that do not significantly deviate from users' current interests given the sparse interactions. Such gradual exploration is guided by a conservative evidential Q-value that prevents recommending totally irrelevant items, causing user frustrations. We theoretically prove that the conservative evidential Q-value provides an uncertainty-aware adjustment of optimism towards the behavior policy that represents users' current interests in an off-policy formulation. Also, we demonstrate that such Q-value estimation produces a lower bound on the actual value of the target policy, providing a conservative view for safe recommendations. We further show that such a conservative view can be incorporated into a policy learning procedure with theoretically guaranteed policy improvement.

The proposed ECQL seamlessly integrates two major components: a sequential state encoder and a *Conservative Evidential Actor-Critic (CEAC)* module. The former primarily focuses on generating the current state of the environment by aggregating the previous state, the current items captured by a sliding window, and the future items from the recommendation. This provides an effective means of dynamic state representation for better future recommendations. Meanwhile, the CEAC module leverages evidential uncertainty to effectively explore the item space to recommend items that potentially align with the user's long-term interest. It encourages learning the optimal policy by maximizing a novel conservative evidential Q-value to make more diverse recommendations that may reflect a long-term interest while keeping a conservative view that does not deviate too much from current interests. The main contribution of this paper is five-fold:

- a novel recommendation model that integrates reinforcement learning with evidential learning to provide uncertainty-aware diverse recommendations that may reflect users' long-term interests,
- evidential uncertainty guided exploitation to maximize information gain to inform model learning,
- conservative off-policy formulation to avoid over-estimation of policy value, leading to low-quality recommendations due to overfitting to sparse training data from limited user interactions,
- a thorough theoretical analysis to justify the desired convergence behavior and recommendation quality that guarantees to avoid risky (or overly optimistic) recommendations.

- seamless integration of a sequential encoder, an actor-critic network, and an evidence network to provide an end-to-end integrated training process.

We conduct extensive experiments over four real-world datasets and compare with state-of-the-art baselines to demonstrate the effectiveness of the proposed model.

## 2 RELATED WORK

**Dynamic and sequential models.** Dynamic recommendation model shifts latent user preference over time to incorporate temporal information. TimeSVD++ (Koren, 2009) considers time-specific factors, which uses additive bias to model user and item-related temporal changes. Gaussian state-space models have been used to introduce time-evolving factors with a one-way Kalman filter (Gultekin & Paisley, 2014). To process implicit data, Sahoo et al. (2012) extended the hidden Markov model and Charlin et al. (2015) further augmented it with the Poisson emission. However, these models capture user evolving preferences, and they are less aware of future interactions and provide recommendations based on fixed strategies. Similarly, sequential models utilize users' historical interactions to capture users' preferences over time. Tang & Wang (2018) utilized a CNN architecture to capture union-level and point-level contributions. Also, Kang & McAuley (2018) leveraged transformer-based user representation to better capture their interest, and Sun et al. (2019) utilized a bidirectional encoder for a sequential recommendation. Similarly, $S^3$-Rec (Zhou et al., 2020) leverages the intrinsic data correlation with mutual information maximization to derive a self-supervised signal to enhance the data representation. Also, CL4SRec (Xie et al., 2022) utilizes contrastive learning to learn the self-supervised signal from the user behavioral data, which helps to extract more meaningful patterns for user representation. However, sequential models neglect long-term users' preferences. The proposed ECQL model aims to fill this critical gap by performing evidence-guided exploration and maximizing the total expected reward.

**RL-based models.** RL-based RS models aim to learn an effective policy to maximize the total expected reward in the long run. The on-policy learning with contextual bandit (Li et al., 2010) and Markov Decision Process (MDP) (Zheng et al., 2018) exploits by interacting with real customers in an online environment. A collaborative contextual bandit algorithm called CoLin (Wu et al., 2016) utilizes graph structure in a collaborative manner. On the other hand, off-policy utilizes Monte Carlo (MC) and temporal-difference (TD) methods to achieve stable and efficient learning with users' history (Farajtabar et al., 2018). Also, Zou et al. (2019) utilized RL to optimize long-term user engagement in recommender systems via Q-network in hierarchical LSTM. Similarly, model-based RL models user-agent interaction via a generative adversarial network (Bai et al., 2019). Pseudo Dyna-Q (Zou et al., 2020) further integrates both direct and indirect RL approaches in a single unified framework without requiring real customer interactions. Recently, SAR (Antaris & Rafailidis, 2021) has leveraged an actor-critic network, where the action is generated as adaptive sequence length to better represent the user's sequential pattern. Similarly, ResAct (Xue et al., 2022) utilizes residual actor-network to reconstruct policy that is close but better than online policy more efficiently in sequential recommendation. However, the above methods utilize random exploration strategies, which are less effective at capturing users' long-term preferences. In contrast, our ECQL utilizes evidence-based uncertainty to systematically explore the item space to maximize the long-term reward. In addition, we encourage an uncertainty-aware conservative view to make safe recommendations due to the sparse RS reward space and rapidly evolving user preferences.

## 3 PRELIMINARIES

**Recommendation Formulation with RL.** We formulate recommendation tasks in an RL setting, where an RL agent interacts with the environment (*i.e.,* users and items) to recommend the next items to a user over time in a sequential order to maximize the cumulative reward. We design this problem as the MDP, which includes a sequence of states, actions, and rewards. More formally, a tuple $(\mathcal{S}, \mathcal{A}, \mathcal{P}, \mathcal{R})$ is defined follows. A state $\mathbf{s}_t = \text{SSE}(\cdot|\mathbf{s}_{t-1}, \mathbf{u}_t) \in \mathcal{S}$ is generated by a sequential state encoder that utilizes previous state $\mathbf{s}_{t-1}$ and current user embedding $\mathbf{u}_t$ which is generated from the concatenation of $N$ items provided by a sliding window and an RL-agent. An action $\mathbf{a}_t \in \mathcal{A}$ is represented as a continuous parameter vector that encodes the user's current and potential preferences and functions as a hidden variable to help recommend top-$N$ items for a user given the current state $\mathbf{s}_t$. The transition probability $\mathcal{P}(\mathbf{s}_{t+1}|\mathbf{s}_t, \mathbf{a}_t)$ quantifies the probability from state $\mathbf{s}_t$ to $\mathbf{s}_{t+1}$ with an action $\mathbf{a}_t$. The RS environment provides an immediate reward $r_t \in \mathcal{R}$ as an RL feedback traditionally based on the recommendation quality in the current time step.

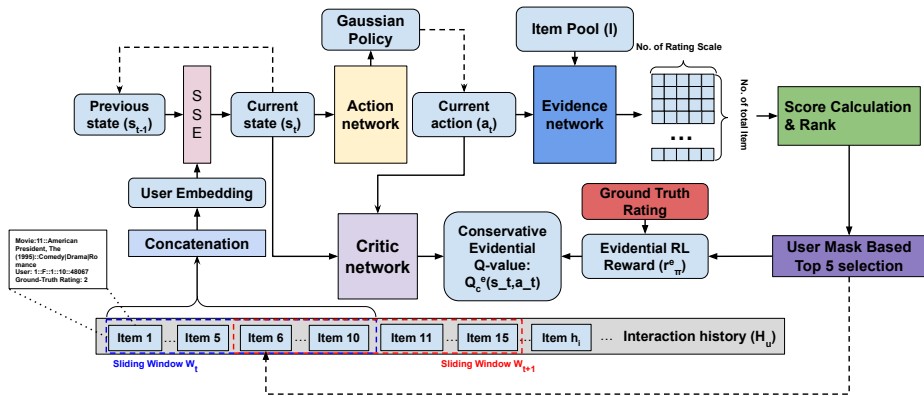

Figure 2: Overview of the ECQL framework

**Uncertainty and Evidential Theory.** Subjective Logic (SL) (Jsang, 2016) is a probabilistic logic that is built upon probability theory and belief theory. It represents uncertainty by introducing vacuity of evidence in its opinion, which is a multinomial random variable $y$ in a $K$-simplex domain $\mathbb{Y}$. This opinion can be equivalently represented by a $K$-dimensional Dirichlet distribution $\text{Dir}(\boldsymbol{p}|\boldsymbol{\alpha})$ where $\boldsymbol{\alpha}$ is a strength vector over $K$ classes and $\mathbf{p} = (p_1, ..., p_K)^\top$ governs a categorical distribution over $\mathbb{Y}$. The term *evidence* $\mathbf{e} = (e_1, \cdots, e_K)^\top$ is a measure of the number of supportive observations from data for each class. It has a fixed relationship $e_k = \alpha_k - 1$ with the Dirichlet strength $\boldsymbol{\alpha}$ given a non-informative prior. Let $e_k$ be the evidence for a class $k$. SL measures different types of second-order uncertainty through evidence, including vacuity, dissonance, and a few others (Josang et al., 2018). In particular, vacuity corresponds to the uncertainty mass of a subjective opinion $\omega$:

$$\texttt{vac}(\omega) = \mathcal{U}(\cdot|\mathbf{e}) = \frac{K}{S}, \quad S = \sum_{k=1}^{K}(e_k + 1). \tag{1}$$

Since vacuity is defined by the lack of evidence (knowledge) in data samples, it provides a natural way to facilitate the exploration.

## 4 EVIDENTIAL CONSERVATIVE Q-LEARNING RS MODEL

**Overview.** We propose an Evidential Conservative Q-learning RS model to perform dynamic recommendations as shown in Figure 2. The model includes a sequential state encoder (SSE) to maintain a dynamic state space with a sliding window $W_t$ which moves along the user's interaction history $H_u$ over time to input new data into the SSE, and a conservative evidential-actor-critic (CEAC) module which functions as an RL agent to explore the item space by introducing the evidence-based uncertainty (*i.e.,* vacuity) into a new off-policy evidential RL setting. By incorporating previous state information, recent items captured by a sliding window, and the recommended items from the RL agent, the sequential encoder generates the current state $\mathbf{s}_t$. This state is further passed to the action network that predicts the mean and variance to form a Gaussian policy distribution. We sample a current action $\mathbf{a}_t$ from the policy distribution that corresponds to the latent preference of the user that simultaneously captures the past (via a previous state), current (through a sliding window) and future interest (through RL exploration). By leveraging the current action and total item embeddings from the Item Pool ($\mathcal{I}$), the evidence network provides the evidence that can be used to form the rating prediction for exploitation while estimating the uncertainty for better exploration. The Q-network (critic) generates a conservative evidential Q-value for conservative policy updates of the action network. Table 4 in Appendix A summarizes the major notations.

### 4.1 ENVIRONMENT SETUP

We start by describing the environment of the proposed evidential RL agent. The environment consists of a buffer containing all the user-interacted item embeddings with ground-truth ratings (*i.e.,* the user's interaction history $H_u$), total item embeddings (*i.e.,* Item Pool $\mathcal{I}$), a sliding window $W_t$ that moves along the interaction history to generate new data input each time into the SSE for dynamic state maintenance, and a recommendation mechanism that specifies a score function to rank items and select top-$N$ of them to recommend. The score function encourages a balance between exploitation

(based on predicted ratings) and exploration (based on evidence-based uncertainty) and is defined as:

$$\text{score}_{u,i} = \widehat{\text{rating}}_{u,i} + \frac{\lambda}{\log(h_i - W_t + 1)} \mathcal{U}_\pi(\cdot|\mathbf{e}_i), \quad i \in H_u, h_i > W_t. \tag{2}$$

where $\lambda \in (0,1)$ balances the rating and the uncertainty, and $\widehat{\text{rating}}_{u,i}$ is the predicted rating for exploitation, $h_i$ is the original appearance position of a recommended item $i$ in the user's interaction history. Given $K$ possible rating classes, the evidence network (introduced later in this section) outputs an evidence vector $\mathbf{e}_i = (e_{i1}, ..., e_{iK})^\top$ for each item $i$. This will allow us to evaluate $\widehat{\text{rating}}_{u,i}$ as $\sum_{k=1}^{K} p_{ik} \times k$ where $p_{ik}$ is rating probability. Meanwhile, vacuity $\mathcal{U}_\pi(i|\mathbf{e}_i)$ for item $i$ can be evaluated through (1) for exploration. $W_t$ indexes the time step reached by the current sliding window, which separates the observed items from the unknown ones. We aim to recommend unknown items that reflect the user's future interests while avoiding too risky recommendations far ahead of the current observation. To this end, we consider the original appearance position $h_i$ and down-weight the information value (captured by vacuity $\mathcal{U}_\pi(i|\mathbf{e}_i)$) of items that are interacted deep into the future. During testing, no down-weight is applied as $h_i$ is unavailable. Based on the ranking score, an RL agent will choose the top-$N$ items to form a list $\mathcal{N}_u$ and recommend them to the user. As feedback to the agent, the user provides the actual rating for each recommended item. Consequently, the evidential reward is

$$r_\pi^e(\mathbf{s}_t, \mathbf{a}_t) = \underbrace{\frac{1}{N} \sum_{i \in \mathcal{N}_u} \text{rating}_{u,i}}_{\mathbf{r}} + \underbrace{\lambda \frac{1}{N} \sum_{i \in \mathcal{N}_u} \mathcal{U}_\pi(\cdot|\mathbf{e}_i)}_{\mathbb{R}}. \tag{3}$$

where $\text{rating}_{u,i}$ is the user assigned ground truth rating. It measures the recommended items' rating as a traditional reward $\mathbf{r}$ balanced with their vacuity predictions as a measure of information gain, denoted as an uncertainty regularizer $\mathbb{R}$.

**Remark:** The novel use of vacuity, which is an evidence-based second-order uncertainty, for exploration in RL, can effectively identify uncertain and informative items (from large item space), indicative of users' long-term interest. In particular, the proposed evidential reward encourages the RL agent to recommend items that the model has the least knowledge (as indicated by a high vacuity). After collecting the user feedback, the RL agent can most effectively gain knowledge of the user preference to make better recommendations in the long run. It should be noted that maximum entropy-based exploration, such as soft-actor-critic (SAC) (Haarnoja et al., 2018), may not reach an optimum policy. It has been shown that a high entropy may imply either high vacuity (lack of evidence) or high dissonance (conflict of strong evidence) (Shi et al., 2020). However, dissonance is not effective for exploration in RS due to its focus on confusing items mostly derived based on the users' current interests. We have also empirically shown this in a qualitative study by demonstrating better recommendation performance than SAC-based exploration in the experiment section.

## 4.2 SEQUENTIAL STATE ENCODER

A specially designed sequential state encoder (SSE) is used to maintain the state space of a dynamic RS environment. In particular, a state $\mathbf{s}_t$ is generated by aggregating three pieces of information: the previous state $\mathbf{s}_{t-1}$, items newly interacted by the user as the sliding window moves, and newly recommended items. By aggregating all this information, the current state can evolve from the previous state by a concatenated user embedding $u_t$ which effectively captures the past preference and future predicted preference of the user. In particular, a sliding window $W_t$ of length $2N$ starts from the beginning of the user's interaction history ($H_u$) at time step 0, then it moves forward along the interaction history $N$ items each time as the newly observed items. The first half $N$ already observed items in the moved window $W_{t+1}$ will be replaced by the top-$N$ recommendation list at last time step $t$, the rest will be the newly observed items in time step $t + 1$. Here, an *item* means an embedding vector that is generated by a pre-trained Word2Vec network to encode item information from raw text descriptions. Then, $\mathbf{s}_t$ is formed by

$$\mathbf{s}_t = \text{SSE}(\mathbf{s}_{t-1}, \mathbf{u}_t). \tag{4}$$

We train the SSE by optimizing action $\mathbf{a}_t$ to maximize the conservative evidential Q-value in the critic network, which gives Equation (5), where $J_\pi(\phi, \omega)$ is the loss objective of the action network.

$$\nabla_\omega J_{\text{SSE}}(\omega) = \nabla_\omega J_\pi(\phi, \omega). \tag{5}$$

### 4.3 Conservative Evidential Actor Critic (CEAC)

CEAC consists of a conservative off-policy training formulation as well as three key networks: *action network*, *critic network*, and *evidence network*, which will be detailed next.

**Conservative off-policy formulation.** RS models aim to leverage offline interactions to predict users' future interests. Since the interaction data is inherently sparse, it may train a RL model that overfits the limited training data, leading to overestimates Q-values of previously unseen interactions. To address the data scarcity and sparse rewards in a typical RS environment, we apply an off-policy learning scheme to promote the reuse of previously collected data and stabilize the training. To further prevent overestimation of the policy value, we extend the conservative Q-learning strategy developed for off-line RL settings (Kumar et al., 2020) and leverage it to penalize the target updated policy $\pi$'s Q-value estimation to avoid over-optimistic predictions on unknown items' ratings and information value as defined in (3). Meanwhile, we increase the Q-value estimate of the behavior policy $\pi_\beta$ that encodes the current knowledge of user interests to avoid the target policy from deviating too much from previous knowledge. These conservative regularizers are balanced with the traditional Bellman training objective using a hyper-parameter $\alpha$. In particular, we run the behavior policy in the RS environment $T$ time steps for one user $u$ as one RL episode. In each time step $t$, RL model collects training tuples $(s_t, a_t, r^e_{\pi,t}, s_{t+1})$ into a replay buffer $D$. We iterate $M$ such RL episodes and then conduct a conservative evidential Q-value $Q^e_c$ evaluation of the current learned policy by minimizing its Q-value while maximizing the Q value of the behavior policy, as given in (6), where $\theta$ is the parameter of critic network to minimize the loss objective $J_{Q^e_c}$:

$$
J_{Q^e_c}(\theta) = \mathbb{E}_{(\mathbf{s}_t,\mathbf{a}_t,r^e_{\pi,t},\mathbf{s}_{t+1})\sim D}\left[\frac{1}{2}(Q(\mathbf{s}_t,\mathbf{a}_t) - \hat{\mathcal{B}}^\pi Q(\mathbf{s}_t,\mathbf{a}_t))^2\right]
$$
$$
+ \alpha\left(\mathbb{E}_{\mathbf{s}_t\sim D,\mathbf{a}_t\sim\pi(\cdot|\mathbf{s}_t)}[Q(\mathbf{s}_t,\mathbf{a}_t)] - \mathbb{E}_{\mathbf{s}_t\sim D,\mathbf{a}_t\sim\pi_\beta(\cdot|\mathbf{s}_t)}[Q(\mathbf{s}_t,\mathbf{a}_t)]\right). \tag{6}
$$

After conservative policy evaluation, we conduct a conservative policy improvement by optimizing the policy towards the optimal conservative evidential Q-value $Q^e_c$ objective as detailed in the action network. After policy improvement, we get a newly learned policy $\pi_{k+1}$ and again alternate between the policy evaluation and improvement steps until convergence. Once we get a converged policy, we update the previous behavior policy $\pi_\beta$ with the newly learned stable policy and begin the next $M$ RL episodes by collecting new training tuples into the replay buffer with the updated behavior policy. After iterating among all users in the training set, we call it an RL epoch. The detailed conservative off-policy formulation is shown in Algorithm 1 of Appendix D.

**Action network.** The action network (or policy network) utilizes the current state $\mathbf{s}_t$ from the offline replay buffer and outputs a policy distribution $\pi(\cdot|\mathbf{s}_t)$, which is modeled as a Gaussian. From this distribution, we sample an action $\mathbf{a}_t$ that is used in the evidence and the critic networks to provide recommendations or direct the policy update. According to our off-policy formulation, we use two separate action networks to represent the currently updated policy and previous behavior policy, respectively. The training of action network is given by

$$
\nabla_\phi J_\pi(\phi) = (-\nabla_{\mathbf{a}_t}Q^e_c(\mathbf{s}_t,\mathbf{a}_t)) \times \nabla_\phi\pi(\mathbf{a}_t|\mathbf{s}_t,\phi). \tag{7}
$$

where $\phi$ is the parameter of the action network to minimize the loss objective $J_\pi(\phi)$.

**Critic network.** The critic network is designed to approximate conservative evidential Q value by utilizing the current state $\mathbf{s}_t$ and action $\mathbf{a}_t$ in a fully connected neural network $Q_\theta(\mathbf{s}_t,\mathbf{a}_t)$. This Q-value judges whether the agent-generated actions match our training requirements. We derived an update formulation (6) for the critic network following the double DQN (Hasselt, 2010) that utilizes two critic networks to stabilize the training process, achieve faster convergence, and provide a better Q-value. Furthermore, the Q-network is optimized with stochastic gradient descent which back-propagates to the action network as well as the sequential encoder in an end-to-end fashion.

**Evidence network.** The evidence network predicts a Dirichlet distribution of class probabilities, which can be considered as an evidence-collection process. The learned evidence $\mathbf{e}_i = (e_{i1}, ..., e_{iK})^\top$ is informative to quantify the predictive uncertainty of recommended items. The network takes action $\mathbf{a}_t$ from the replay buffer and item pool $\mathcal{I}$ to provide class-level evidence. Then, the probability of rating $k$ is $p_{ik} = (e_{ik} + 1)/S_i$. To train the evidence network, we define a standard evidential loss (8) by utilizing the MSE loss between rating class probability $p_{ik}$ and the one-hot ground truth label

$\mathbf{y}_i$, in which $y_{ik} = 1$ if $k$ is the correct rating, otherwise $y_{ik} = 0$:

$$J_{Evi}(\psi) = \sum_{i \in H_u} \sum_{k=1}^{K} (y_{ik} - p_{ik})^2 + \frac{p_{ik}(1 - p_{ik})}{S_i + 1}. \tag{8}$$

We update the network by back-propagating the evidential loss $J_{Evi}(\psi)$ with its parameters $\psi$.

## 4.4 Derivation of Conservative Evidential Policy Iteration

We first highlight some important concept differences resulting from our novel evidential conservative setting: 1) $\pi$ is actually an evidential policy governed by an evidential reward $r_\pi^e$, and 2) $\pi_\beta$ is the true behavior policy directly available from the off-policy setting without any simulation from offline data. Resulting of these two key symbol differences, the empirical Bellman operator $\hat{\mathcal{B}}^\pi$ should also be an evidential Bellman operator, where $\pi$ is an updated evidential policy. However, to leverage the existing theorems in relevant RL literature which are derived under the traditional RL concepts, we make evidential terms explicit by restoring: 1) the evidential reward $r_\pi^e$ to $r + \mathcal{R}$, where $\mathcal{R}$ is the vacuity term in our case, and 2) evidential policy $\pi$ to non-evidential target policy $\bar{\pi}$. As a result, the empirical Bellman operator returns to its non-evidential version $\hat{\mathcal{B}}^{\bar{\pi}}$ and $\pi_\beta$ return back to non-evidence guided behavior policy $\bar{\pi}_\beta$. Based on these newly defined concepts, we first prove a conservative evidential policy evaluation with Q-value updated in (6). In Appendix C.2, we further show that such policy evaluation leads to a $\zeta$-safe policy improvement over the behavior policy. Then, we disclose its intrinsic relationship with an uncertainty-aware adjustment of optimism towards the previous behavior policy.

**Lemma 1 (Conservative Evidential Policy Evaluation)** *Given a policy $\bar{\pi}$ and its conservative evidential Q value estimation $Q_c^e$ updated using (6), the expected conservative state value estimation $\hat{V}^{\bar{\pi}}(\mathbf{s})$ always lower-bounds the actual state value $V^{\bar{\pi}}(\mathbf{s})$ for any state $s$ when the balancing factor*

$$\alpha \geq \frac{C_{r,T,\delta} R_{\max}}{1 - \gamma} \cdot \max_{\mathbf{s} \in D} \frac{1}{\sqrt{|D(\mathbf{s})|}} \left[ \sum_{\mathbf{a}} \frac{(\bar{\pi}(\mathbf{a}|\mathbf{s}) - \pi_\beta(\mathbf{a}|\mathbf{s}))^2}{\bar{\pi}_\beta(\mathbf{a}|\mathbf{s})} \right]^{-1}. \tag{9}$$

**Remark:** Due to the interaction between the evidence-based uncertainty in the evidential reward defined in (2), it leads to a more strict constraint (through the balancing parameter $\alpha$) in conservative policy evaluation to avoid risky recommendations when performing evidence based exploration. This is one critical difference from standard conservative Q-learning (Kumar et al., 2020), which does not consider evidential exploration. The detailed notation definition and proof are in Appendix C.1.

**Theorem 2 (Uncertainty Aware Optimism Adjustment)** *The conservative evidential Q-value update in (6) has an intrinsic relationship with an uncertainty-aware adjustment of optimism towards the behavior policy by representing the evidential reward $r_\pi^e$ with a traditional reward $r$ added by an uncertainty regularizer $\mathbb{R}$, thus leading to a normal conservative Q-value update:*

$$\hat{Q}^{k+1} \longleftarrow \min_{\hat{Q}^k} \frac{1}{2} \mathbb{E}_{\mathbf{s},\mathbf{a},r,\mathbf{s}' \sim D} \left[ \left( \hat{Q}^k(\mathbf{a}, \mathbf{s}) - \hat{\mathcal{B}}^{\bar{\pi}} \hat{Q}^k(\mathbf{a}, \mathbf{s}) \right)^2 \right] + \alpha \mathbb{E}_{\mathbf{s} \sim D, \mathbf{a} \sim \bar{\pi}(\cdot|\mathbf{s})} [\hat{Q}^k(\mathbf{a}, \mathbf{s})]$$
$$- \alpha \left( \mathbb{E}_{\mathbf{s} \sim D, \mathbf{a} \sim \bar{\pi}_\beta(\cdot|\mathbf{s})} (1 + \mathbb{R}) [\hat{Q}^k(\mathbf{a}, \mathbf{s})] \right) + \mathcal{C}. \tag{10}$$

*where $\mathcal{C}$ is a constant.*

**Remark:** The theorem manifests another novel interplay between the evidential uncertainty and the conservative policy update. From (10), we observe that the uncertainty regularizer $\mathbb{R}$ serves as an importance weight for the Q-value estimation of the behavior policy $\bar{\pi}_\beta$. This observation provides us with an elegant interpretation of the proposed ECQL algorithm: by adding the evidence-based uncertainty measure, we are actually adjusting the optimism of the previously learned behavior policy $\bar{\pi}_\beta$ according to the information gain (quantified by the vacuity) of a chosen action. If an action generated by a behavior policy leads to low information gain in the recommendation list (*i.e.,* low average vacuity), then the importance weight for the Q-value of such behavior policy is lower and the ECQL algorithm will discourage the RL agent to keep imitating the behavior policy. We leave the detailed derivation in Appendix C.3.

Table 2: Performance of Recommendation (average P@N and nDCG@N)

| Category | Model | MovieLens-1M | | MovieLens-100K | | Netflix | | Yahoo! Music | |
|---|---|---|---|---|---|---|---|---|---|
| | | P@5 | nDCG@5 | P@5 | nDCG@5 | P@5 | nDCG@5 | P@5 | nDCG@5 |
| Sequential | CASER | 0.5762 | 0.4613 | 0.5434 | 0.4428 | 0.5633 | 0.4532 | 0.5745 | 0.4315 |
| | SASRec | 0.6058 | 0.4862 | 0.5624 | 0.4515 | 0.5958 | 0.4621 | 0.5826 | 0.4422 |
| | BERT4Rec | 0.6122 | 0.4957 | 0.5834 | 0.4855 | 0.5996 | 0.4667 | 0.5901 | 0.4522 |
| | Seq2Seq | 0.5818 | 0.4752 | 0.5579 | 0.4614 | 0.5648 | 0.4554 | 0.5762 | 0.4332 |
| | $S^3$-Rec | 0.6108 | 0.4926 | 0.5792 | 0.4767 | 0.5884 | 0.4602 | 0.5786 | 0.4358 |
| | CL4SRec | 0.6135 | 0.4952 | 0.5813 | 0.4781 | 0.5902 | 0.4688 | 0.5841 | 0.4423 |
| Reinforce | $\epsilon$-greedy | 0.5977 | 0.4834 | 0.5580 | 0.4556 | 0.5850 | 0.4765 | 0.5909 | 0.4812 |
| | DRN | 0.6057 | 0.5199 | 0.6154 | 0.5268 | 0.5826 | 0.4720 | 0.6085 | 0.5121 |
| | LIRD | 0.6238 | 0.5332 | 0.6137 | 0.5222 | 0.6134 | 0.5214 | 0.6193 | 0.5238 |
| | CoLin | 0.6162 | 0.5216 | 0.6247 | 0.5285 | 0.5869 | 0.4782 | 0.6112 | 0.5194 |
| Proposed | **ECQL** | **0.6313** | **0.5365** | **0.6379** | **0.5386** | **0.6336** | **0.5372** | **0.6232** | **0.5330** |

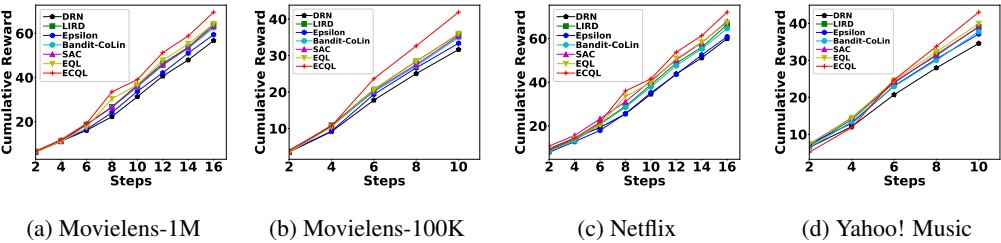

(a) Movielens-1M  (b) Movielens-100K  (c) Netflix  (d) Yahoo! Music

Figure 3: Average cumulative reward for ECQL and other RL-baselines across different time steps

## 5 EXPERIMENTS

We conduct extensive experiments on four real-world datasets that contain explicit ratings: *Movielens-1M*, *Movielens-100K*, *Netflix*, and *Yahoo! Music*. For baseline comparisons, we use **sequential** models: CASER (Tang & Wang, 2018), SASRec (Kang & McAuley, 2018), BERT4Rec (Sun et al., 2019), Seq2Seq (Ma et al., 2020), $S^3$-Rec (Zhou et al., 2020), CL4SRec (Xie et al., 2022); and **RL-based** models: $\epsilon$-greedy (Zhao et al., 2013), DRN (Zheng et al., 2018), LIRD (Zhao et al., 2017), CoLin (Wu et al., 2016). We use two standard metrics: **Precision@N** and **nDCG@N** to measure the average recommendation performance across all time steps in different test users. We also use **Cumulative Reward** as a measure of the average rating of recommendation for the RL-based methods comparison. For any particular user, we use **Positive Count**, and **Genre Count** directly to measure the qualitative performance. Further details about datasets, metrics, settings, and baseline are in Appendix E.

### 5.1 RECOMMENDATION PERFORMANCE COMPARISON

**Recommendation performance.** Table 2 summarizes the recommendation performance from all models. The proposed model benefits from both the sequential state encoder and CEAC module so that it provides better results in all datasets. Sequential models achieve less ideal performance due to their focus on short-term user interest and inability to provide long-run or future preference. RL methods have shown a clear advantage due to their focus on maximizing expected long-term rewards. By leveraging the vacuity guided exploration, ECQL achieves the best performance among all RL based models. Further details about datasets, metrics, settings, and baseline are in Appendix E.

**Cumulative reward for RL-based methods.** We report the cumulative reward, which measures recommendation performance on test users. We plot the average cumulative rewards for the proposed ECQL model and baseline RL-models in Figure 3. As can be seen, the cumulative rewards for ECQL and RL-based model in the initial epochs are quite close. But in later epochs, ECQL clearly outperforms the other baselines. This is because the model explores more effectively during the training process to enhance the knowledge of the model. Furthermore, we include one classic factor-machine based DeepFM (Guo et al., 2017), DL based DCNv2 (Wang et al., 2021) and RL-based REINFORCE (Chen et al., 2019a) in the Appendix to demonstrate our proposed model's superiority over all these baselines. We attribute this improved performance to vacuity guided effective exploration as well as conservative learning to make accurate recommendations by conducting ablated qualitative studies, as detailed in Section 5.2.

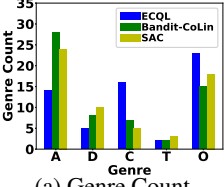 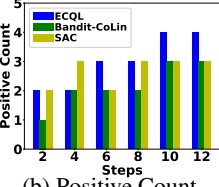 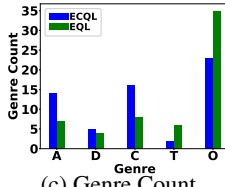 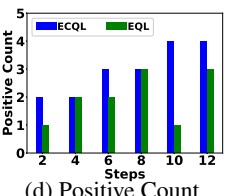

|     (a) Genre Count     |     (b) Positive Count     |     (c) Genre Count     |     (d) Positive Count     |

Figure 4: Genre and Positive Count comparisons with RL models using bandit-based (CoLin) and entropy-based (SAC) explorations (a-b) as well as EQL without conservative learning (c-d)

Table 3: Recommended movies for UserID: 4967

| Model | Movies | Movie Genre | Vacuity |
|-------|--------|-------------|---------|
|       | Kids of the Round Table (1995) | **Adventure,Fantasy** | 0.12 |
|       | Postino, Il (The Postman) (1994) | **Adventure, Romance** | 0.11 |
| ECQL  | How to Make an American Quilt (1995) | **Drama** | 0.14 |
|       | Pocahontas (1995) | **Musical** | 0.12 |
|       | Three Lives and Only One Death (1996) | Comedy | 0.22 |
|       | Karate Kid, Part II, The (1986) | **Fantasy** | 0.07 |
|       | Return of the Pink Panther, The (1974) | Comedy | 0.12 |
| SAC   | Lawnmower Man 2: Beyond Cyberspace (1996) | Sci-Fi,Thriller | 0.09 |
|       | Young Sherlock Holmes (1985) | **Adventure** | 0.11 |
|       | Love in Bloom (1935) | **Romance** | 0.09 |
|       | Ruling Class, The (1972) | Tragedy | 0.14 |
|       | Private Benjamin (1980) | Comedy | 0.13 |
| EQL   | Mighty Joe Young (1998) | **Adventure** | 0.08 |
|       | Christmas Vacation (1989) | Action | 0.14 |
|       | Father of the Bride Part II (1995) | Sci-Fi | 0.14 |

## 5.2 QUALITATIVE STUDY

**Impact of vacuity for exploration.** We conduct a qualitative analysis to show the advantage of using evidence-based uncertainty (*i.e.,* vacuity) for RL exploration when compared with other two competitive baselines: entropy-guided exploration as in the soft actor-critic (SAC) (Haarnoja et al., 2018) and a contextual bandit algorithm (CoLin) (Wu et al., 2016). We select a random test user (ID:4967) from the Movielens-1M dataset and show the genre counts and positive counts of recommended items in Figure 4. At the initial few steps, SAC has more positive counts but is less effective in exploration. The proposed ECQL is able to explore more informative items (evidenced by more diverse genres). In later steps, ECQL consistently outperforms both SAC and CoLin in positive counts due to the better utilization of evidence-based uncertainty to discover more informative future items which could reflect the user's long-term preference, as shown in the plot 4b. Table 3 shows the predicted vacuity for each recommended item in step 10. The overall higher vacuity scores indicate that ECQL recommends more items that are currently unknown to the users, which is instrumental to explore their long-term interests. ECQL also explores more diverse genres and identifies four out of five important items (genre types in bold) that are positive in ground-truth ratings. Benefiting from better exploration, ECQL eventually achieves a much better cumulative reward compared to SAC, Bandit-CoLin as shown in Figure 3. More detailed discussions about how to control the exploration through hyper-parameter $\lambda$ are included in Appendix E.3.

**Impact of conservative learning.** The right two plots in Figure 4 show that comparing to EQL, which lacks a conservative Q-learning constraint, our proposed model achieves much higher positive counts across different steps while exploring diverse genres. Because of the sparse reward space and limited training data, making reasonable and safer recommendations based on current confirmed knowledge is important to maintain a user base. As shown in Table 3, EQL explores a wide range of different types of movies, but only one of them is the positive movie reflecting user's true interests.

## 6 CONCLUSION

In this paper, we propose a novel evidential conservative Q-learning framework for dynamic recommendations. The proposed ECQL framework learns an effective and safe recommendation policy by integrating both the evidence-based exploration and conservative learning. ECQL integrates a customized SSE to generate the current state that accurately captures user interest and a conservative evidential-actor-critic module which functions as an RL agent to perform evidence-based exploration and trained through conservative evidential Q-learning in an off-policy formulation. We theoretically prove that the conservative evidential Q-value provides an uncertainty-aware adjustment of optimism towards the behavior policy and lower-bounds the actual value of the target policy.

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

# Appendix

ORGANIZATION OF APPENDIX

In this Appendix, we first summarize the major mathematical notations in Appendix A. We give an overview of representative static recommendation models in Appendix B. We then present the proofs of lemmas and theorems in Appendix C. We show the detailed ECQL algorithm in Appendix D. We present the details of the datasets, experimental setting and baseline models in Appendix E. Further, we include some additional comparison results in Appendix E.2 and conduct an ablation study as well as a qualitative analysis in Appendix E.3 and E.4. Broader impact, limitation and future extension of our work are discussed in Appendix F. The link to the source code is given in Appendix G.

## A SUMMARY OF NOTATIONS

Table 4: Summary of Notations

| Symbol Group | Notation | Description |
|---|---|---|
| ECQL | $T, M$ | step size or length of episode and episode size for ECQL training |
| | $u, i, h_i$ | user (episode), item and item's position indices |
| | $\mathbf{u}_t$ | user $u$'s embedding at time step t |
| | $\mathbf{s}_t, \mathbf{a}_t$ | state and action at time t |
| | $e_{ik}, p_{ik}$ | evidence and evidence-based probability on rating class $k$ for item $i$ |
| | $y_i$ | one-hot rating label on item $i$ |
| | $\phi, \psi$ | parameters of action and evidence networks |
| | $\theta, \omega$ | parameters of critic and SSE networks |
| | $\pi, Q(\mathbf{s}_t, \mathbf{a}_t)$ | recommendation policy or action network, Q value function or critic network |
| | $\widehat{rating}_{u,i}, rating_{u,i}$ | predicted and actual rating for user $u$ on item $i$ |
| | $score_{u,i}$ | evidential score for user $u$ on item $i$ |
| | $\tau, \lambda$ | rating threshold and balance hyper-parameter for exploitation and exploration |
| | $\gamma$ | RL discount factor for cumulative reward |
| | $\mathcal{U}_\pi(\cdot|\mathbf{e}_i)$ | item $i$'s evidence-based uncertainty |
| | $r_\pi^e(\mathbf{s}_t, \mathbf{a}_t), Q^e(\mathbf{s}_t, \mathbf{a}_t)$ | evidential reward and Q value |
| Environment | $\mathcal{P}(\mathbf{s}'|\mathbf{s}, \mathbf{a})$ | state transition probability |
| | $K$ | the number of rating class |
| | $W^l, N$ | the number of interacted items in the sliding window and recommended items in the recommendation list in each time step |
| | $\mathcal{N}_u$ | top-$N$ recommended items list for user $u$ |
| | $H_u, \mathcal{I}$ | user $u$'s interaction history and item pool |
| Theoretical Results | $C_{r,T,\delta}, C_{r,\delta}, C_{T,\delta}$ | constants dependent on the concentration properties (variance) of evidential reward $r_\pi^e(\mathbf{a}, \mathbf{s})$ and/or state transition matrix $T(\mathbf{s}'|\mathbf{s}, \mathbf{a})$ |
| | $\alpha, \delta, \zeta$ | hyper-parameters controlling conservatism, high probability and safe policy improvement guarantee. |
| | $J(\bar{\pi}, \hat{M}), J(\bar{\pi}, M)$ | discounted return of a policy $\bar{\pi}$ in the empirical MDP, $\hat{M}$ and actual MDP, M. |
| | $|D|, |A|, d_{\hat{M}}^{\bar{\pi}}$ | magnitude of state and action space, state distribution from empirical MDP $\hat{M}$(offline dataset) |
| | $D_{ECQL}(\bar{\pi}, \bar{\pi}_\beta)$ | ECQL regularizer detailed in C.1. |

# B  ADDITIONAL RELATED WORK

To provide a complete review of the existing literature on recommender systems, this section gives an overview of representative static recommendation models that complements the dynamic, sequential, and RL-based models discussed in the main paper.

**Static recommendation models.** Matrix Factorization (MF) leverages user and item latent factors to infer user preferences (Koren et al., 2009; Funk, 2006; Koren, 2008). MF is further extended with Bayesian Personalized Ranking (BPR) (Rendle et al., 2012) and Factorization Machine (FM) (Rendle, 2010). Recently, deep learning-based recommender systems (Cheng et al., 2016; Guo et al., 2017) have achieved impressive performance. DeepFM (Guo et al., 2017) integrates traditional FM and deep learning to learn low- and high-order feature interactions. The wide and deep networks are jointly trained in (Cheng et al., 2016) for better memorization and generalization. In graph-based methods (Berg et al., 2017), users and items are represented as a bipartite graph and links are predicted to provide recommendations. Similarly, Neural Graph Collaborative Filtering (Wang et al., 2019) explicitly encodes the collaborative signal via high-order connectivities in the user-item bipartite graph via embedding propagation.

# C  PROOFS OF THEORETICAL RESULTS

In this section, we provide proofs of lemma 1, theorem 2 and $\zeta$-safe policy improvement guarantees over behavior policy by iterating conservative policy evaluation (6) and update (7).

## C.1  PROOF OF LEMMA 1

**Proof overview.** We first introduce some important notations that will be used in the proof of the Lemma. Our proof shows that conservative Q-value estimation updated step by step using (12) leads to a lower-bounded state-marginal value estimation compared to its previous iterate. Then, we compute the fixed point of such iteration to conclude that the converged state value estimation $\hat{V}^{\bar{\pi}}(\mathbf{s})$ lower-bounds the actual state value $V^{\bar{\pi}}(\mathbf{s})$. We first conduct the whole derivation in tabular Q-learning setting where state transition is finite and precise (Auer & Ortner, 2006), then we incorporate the sampling error to make it a complete RL process.

**Notations.** Let $k \in \mathbb{N}$ denote an Q-iteration of policy evaluation. In an iteration $k$, the objective in (6) is optimized using the Bellman backup (*i.e.,* $\hat{\mathcal{B}}^{\bar{\pi}}\hat{Q}^{k-1}$) as the target value which is given as Equation (11) by following a double DQN training fashion, where $\mathcal{R}$ is the evidential regularizer, $\gamma$ is the discounted factor, $\tilde{Q}$ is the target Q-network representing a stable evaluation and $\hat{Q}$ is the Q-network representing the current learned policy's Q-value. Let $Q^k$ denote the true, tabular Q-function iterate in the actual MDP $M$, and $Q^k$ is estimated using the empirical Bellman operator $\hat{\mathcal{B}}$ as: $Q^k \leftarrow \hat{\mathcal{B}}^{\bar{\pi}}\hat{Q}^k$ (for policy evaluation). We redefine $Q_c^e$ in the main paper as $Q^k$ when $k \to \infty$ (*i.e.,* the $k$-th Q-function iterate reaches convergence). We emphasize that $\bar{\pi}_{\beta}$ is the actual behavior policy directly available from last policy update, $\alpha$ is the hyper-parameter to guarantee conservatism, and $\hat{V}^k$ denotes the state value estimation, which is the expectation of corresponding Q-value under a given policy $\bar{\pi}$: $\hat{V}^k := \mathbb{E}_{\mathbf{a}\sim\bar{\pi}(\mathbf{a}|\mathbf{s})}[\hat{Q}^k(\mathbf{a},\mathbf{s})]$.

$$\hat{\mathcal{B}}^{\bar{\pi}}\hat{Q}^{k-1}(\mathbf{s}_t,\mathbf{a}_t) = \mathbb{E}_{\mathbf{s}_t,\mathbf{a}_t,\mathbf{s}_{t+1}\sim D,\mathbf{a}_{t+1}\sim\bar{\pi}}[(r(\mathbf{s}_t,\mathbf{a}_t)$$
$$+ \gamma \times \min\{\tilde{Q}^{k-1}(\mathbf{s}_{t+1},\mathbf{a}_{t+1}),\hat{Q}^{k-1}(\mathbf{s}_{t+1},\mathbf{a}_{t+1})\}].  \quad (11)$$

**Assumptions.** Following (Osband et al., 2016; O' Donoghue, 2021; Auer et al., 2008), we use concentration properties of $\hat{\mathcal{B}}^{\bar{\pi}}$ to control the sampling error. Formally, for all $\mathbf{s},\mathbf{a} \in D$, with probability $\geq 1-\delta$, there exists a constant $C_{r,T,\delta}$ dependent on the concentration properties (variance) of reward $r(\mathbf{s},\mathbf{a})$ and state transition matrix $T(\mathbf{s}'|\mathbf{s},\mathbf{a})$ that satisfies $|\hat{\mathcal{B}}^{\bar{\pi}} - \mathcal{B}^{\bar{\pi}}|(\mathbf{s},\mathbf{a}) \leq \frac{C_{r,T,\delta}}{|\sqrt{D(\mathbf{s},\mathbf{a})}|}$, where $\delta \in (0,1)$. $\frac{1}{|\sqrt{D(\mathbf{s},\mathbf{a})}|}$ denotes a vector of size $|S||A|$ containing square root inverse counts for each state-action pair, except when $D(\mathbf{s},\mathbf{a}) = 0$, in which case the corresponding entry is a very large but finite value $\delta \geq \frac{2R_{\max}}{1-\gamma}$.

**Proof.** In the tabular setting, we can set the derivative of the modified objective in (6) to 0, and compute the Q-function update induced in the exact, tabular setting, thus giving (12) as one optimization step (this assumes $\hat{\mathcal{B}}^{\bar{\pi}} = \mathcal{B}^{\bar{\pi}}$).

$$\forall \mathbf{s}, \mathbf{a} \quad \hat{Q}^k(\mathbf{a}, \mathbf{s}) = \mathcal{B}^{\bar{\pi}}\hat{Q}^{k-1}(\mathbf{a}, \mathbf{s}) + \tilde{\mathcal{R}} - \alpha \left[ \frac{\bar{\pi}(\mathbf{a}|\mathbf{s})}{\bar{\pi}_\beta(\mathbf{a}|\mathbf{s})} - 1 \right]. \tag{12}$$

Where $\tilde{\mathcal{R}}$ is the expectation of the evidential regularizer under the off-policy state-action distribution $D$. Note that for state-action pairs $(\mathbf{a}, \mathbf{s})$ such that, $\bar{\pi}(\mathbf{a}|\mathbf{s}) < \bar{\pi}_\beta(\mathbf{a}|\mathbf{s})$, we are in fact adding a positive quantity, $1 - \frac{\bar{\pi}(\mathbf{a}|\mathbf{s})}{\bar{\pi}_\beta(\mathbf{a}|\mathbf{s})}$. To the Q-function obtained, a point-wise lower bound is not guaranteed, *i.e.,* $\forall \mathbf{s}, \mathbf{a}, s.t. \hat{Q}^k(\mathbf{a}, \mathbf{s}) \leq \hat{Q}^{k-1}(\mathbf{a}, \mathbf{s})$. However, we show that on the other hand the expected state value of the estimated Q-function, *i.e.,* $\hat{V}^k$ always lower-bounds its previous iterate, since:

$$\hat{V}^k(\mathbf{s}) := \mathbb{E}_{\mathbf{a} \sim \bar{\pi}(\mathbf{a}|\mathbf{s})}\left[ \hat{Q}^k(\mathbf{a}, \mathbf{s}) \right] = \mathcal{B}^{\bar{\pi}}\hat{V}^{k-1}(\mathbf{s}) - \alpha \mathbb{E}_{\mathbf{a} \sim \bar{\pi}(\mathbf{a}|\mathbf{s})}\left[ \frac{\bar{\pi}(\mathbf{a}|\mathbf{s})}{\bar{\pi}_\beta(\mathbf{a}|\mathbf{s})} - (1 + \frac{\tilde{\mathcal{R}}}{\alpha}) \right]. \tag{13}$$

The value of the policy $\hat{V}^k$ is underestimated comparing to $\hat{V}^{k-1}$, since we can show that the evidential term $D_{ECQL}(\mathbf{s}) := \sum_{\mathbf{a}} \bar{\pi}(\mathbf{a}|\mathbf{s}) \left[ \frac{\bar{\pi}(\mathbf{a}|\mathbf{s})}{\bar{\pi}_\beta(\mathbf{a}|\mathbf{s})} - (1 + \frac{\tilde{\mathcal{R}}}{\alpha}) \right]$ is always positive under the condition that $\alpha \geq [\sum_{\mathbf{a}} \frac{(\bar{\pi}(\mathbf{a}|\mathbf{s}) - \bar{\pi}_\beta(\mathbf{a}|\mathbf{s}))^2}{\bar{\pi}_\beta(\mathbf{a}|\mathbf{s})}]^{-1}$, which is the evidential constraint on conservative parameter $\alpha$ to guarantee underestimation in each value update. To note this, we present the following derivation:

$$\begin{aligned}
D_{ECQL}(\mathbf{s}) &:= \sum_{\mathbf{a}} \bar{\pi}(\mathbf{a}|\mathbf{s}) \left[ \frac{\bar{\pi}(\mathbf{a}|\mathbf{s})}{\bar{\pi}_\beta(\mathbf{a}|\mathbf{s})} - (1 + \frac{\tilde{\mathcal{R}}}{\alpha}) \right] \\
&= \sum_{\mathbf{a}} (\bar{\pi}(\mathbf{a}|\mathbf{s}) - \bar{\pi}_\beta(\mathbf{a}|\mathbf{s}) + \bar{\pi}_\beta(\mathbf{a}|\mathbf{s})) \left[ \frac{\bar{\pi}(\mathbf{a}|\mathbf{s})}{\bar{\pi}_\beta(\mathbf{a}|\mathbf{s})} - (1 + \frac{\tilde{\mathcal{R}}}{\alpha}) \right] \\
&= \sum_a (\bar{\pi}(\mathbf{a}|\mathbf{s}) - \bar{\pi}_\beta(\mathbf{a}|\mathbf{s})) \left[ \frac{\bar{\pi}(\mathbf{a}|\mathbf{s}) - \bar{\pi}_\beta(\mathbf{a}|\mathbf{s})}{\bar{\pi}_\beta(\mathbf{a}|\mathbf{s})} \right] + \sum_{\mathbf{a}} \bar{\pi}_\beta(\mathbf{a}|\mathbf{s}) \left[ \frac{\bar{\pi}(\mathbf{a}|\mathbf{s})}{\bar{\pi}_\beta(\mathbf{a}|\mathbf{s})} - (1 + \frac{\tilde{\mathcal{R}}}{\alpha}) \right] \\
&= \sum_{\mathbf{a}} \underbrace{\left[ \frac{(\bar{\pi}(\mathbf{a}|\mathbf{s}) - \bar{\pi}_\beta(\mathbf{a}|\mathbf{s}))^2}{\bar{\pi}_\beta(\mathbf{a}|\mathbf{s})} \right]}_{\geq 0} - \frac{\tilde{\mathcal{R}}}{\alpha} \geq 0, \text{(R is an uncertainty regularity between 0 to 1)}
\end{aligned}$$

which requires $\alpha \geq \dfrac{1}{\left[ \sum_{\mathbf{a}} \frac{(\bar{\pi}(\mathbf{a}|\mathbf{s}) - \bar{\pi}_\beta(\mathbf{a}|\mathbf{s}))^2}{\bar{\pi}_\beta(\mathbf{a}|\mathbf{s})} \right]} \geq \dfrac{\tilde{\mathcal{R}}}{\left[ \sum_{\mathbf{a}} \frac{(\bar{\pi}(\mathbf{a}|\mathbf{s}) - \bar{\pi}_\beta(\mathbf{a}|\mathbf{s}))^2}{\bar{\pi}_\beta(\mathbf{a}|\mathbf{s})} \right]}. \tag{14}$

The above derivation implies that each value iterate incurs some underestimation $\hat{V}^k(\mathbf{s}) \leq \hat{V}^{k-1}(\mathbf{s})$ when conservative hyper-parameter $\alpha$ satisfies some constraints. Now we compute the fixed point of value iteration (13) as the equivalency to $\hat{V}^k(\mathbf{s})$ when $k \to \infty$ and get the following estimated policy value $\hat{V}^{\bar{\pi}}(\mathbf{s})$ which shows a clear lower bound to actual state value function $V^{\bar{\pi}}(\mathbf{s})$, given that the simulation error $\left[ (I - \gamma T^{\bar{\pi}})^{-1} \mathbb{E}_{\bar{\pi}} \frac{\hat{\pi}_\beta}{\bar{\pi}_\beta} \right]$ can be avoided because of our off-policy setting:

$$\hat{V}^{\bar{\pi}}(\mathbf{s}) = V^{\bar{\pi}}(\mathbf{s}) - \alpha \left[ \mathcal{E}_{\bar{\pi}} \left[ \frac{\pi}{\bar{\pi}_\beta} - (1 + \frac{\tilde{\mathcal{R}}}{\alpha}) \right] \right]. \tag{15}$$

Then, by further incorporating the sampling error $\left[ (I - \gamma T^{\bar{\pi}})^{-1} \frac{C_{r,T,\delta}R_{\max}}{1 - \gamma\sqrt{|D|}} \right](\mathbf{s})$, we get the following full evidential constraint for $\alpha$, which is a more strict constraint compared to CQL:

$$\alpha \geq \frac{C_{r,T,\delta}R_{\max}}{1 - \gamma} \cdot \max_{\mathbf{s} \in D} \frac{1}{\sqrt{|D(\mathbf{s})|}} \left[ \sum_{\mathbf{a}} \frac{(\bar{\pi}(\mathbf{a}|\mathbf{s}) - \bar{\pi}_\beta(\mathbf{a}|\mathbf{s}))^2}{\bar{\pi}_\beta(\mathbf{a}|\mathbf{s})} \right]^{-1}.$$

## C.2 PROOF OF CONSERVATIVE POLICY IMPROVEMENT

**Lemma 3 (Conservative Policy Improvement)** *Given an optimal policy $\pi_*$ that is the fixed point under action optimization using Equation (7), then the policy $\pi^*(\mathbf{a}|\mathbf{s})$ is a $\zeta$-safe policy improvement over behavior policy $\pi_\beta$ in the actual MDP $M$. The expected discounted return attained by a policy $\pi^*$ in the actual underlying MDP $M$, i.e., $J(\pi^*, M)$, is guaranteed to be higher than that attained by its behavior policy $\pi_\beta$ with a lowest $\zeta$ bound. Formally, we represent it as $J(\pi^*, M) \geq J(\pi_\beta, M) - \zeta$ with a high probability $1 - \delta$ where $\zeta$ is given as:*

$$\zeta = 2\left(\frac{C_{r,\delta}}{1-\gamma} + \frac{\gamma R_{max} C_{T,\delta}}{(1-\gamma)^2}\right) \mathbb{E}_{\mathbf{s} \sim d_{\hat{M}}^{\pi}(\mathbf{s})}\left[\frac{\sqrt{|A|}}{\sqrt{|D(\mathbf{s})|}}\sqrt{D_{ECQL}(\bar{\pi}, \bar{\pi}_\beta)(\mathbf{s}) + 1}\right]$$
$$- \alpha \frac{1}{1-\gamma}\mathbb{E}_{\mathbf{s} \sim d_{\hat{M}}^{\pi^*}(\mathbf{s})}[D_{ECQL}(\pi^*, \bar{\pi}_\beta)(\mathbf{s})].$$

where $\delta$ is a high probability control related to $\zeta$, $d_{\hat{M}}^{\pi}$ is the state distribution from the empirical MDP $\hat{M}$. $|A|, |D|$ is the magnitude (norm) of action and state spaces. $C_{r,\delta}$ and $C_{T,\delta}$ are constants dependent on the concentration properties (variance) of evidential reward $r_\pi^e(\mathbf{a}, \mathbf{s})$ and state transition matrix $T(\mathbf{s}'|\mathbf{s}, \mathbf{a})$, respectively. $\gamma$ is the discounted factor. $D_{ECQL}$ is an evidential conservative factor ensuring the conservatism in policy evaluation.

**Proof.** In this section, we first show that this policy improvement procedure defined in Equation (7) actually optimizes a penalized RL objective $J(\pi, \hat{M}) - \alpha \frac{1}{1-\gamma}\mathbb{E}_{\mathbf{s} \sim d_{\hat{M}}^{\pi}(\mathbf{s})}[D_{ECQL}(\bar{\pi}, \bar{\pi}_\beta)(\mathbf{s})]$ using Lemma D.3.1 following (Kumar et al., 2020), where $J(\pi, \hat{M})$ is the empirical discounted return of policy $\pi$ in empirical MDP $\hat{M}$, $D_{ECQL}$ is the conservatism term given in (14), and then we relate the performance of $\pi^*(\mathbf{a}|\mathbf{s})$ updated with this penalized RL objective, to the performance of itself and its behavior policy $\bar{\pi}_\beta$ in the actual MDP $M$ under the non-evidential settings so that we can leverage existing Theorem D.4 proposed by Kumar et al. (Kumar et al., 2020), thus gives:

$$J(\pi^*, M) \geq J(\bar{\pi}_\beta, M) - 2\left(\frac{C_{r,\delta}}{1-\gamma} + \frac{\gamma R_{max} C_{T,\delta}}{(1-\gamma)^2}\right) \mathbb{E}_{\mathbf{s} \sim d_{\hat{M}}^{\pi^*}(\mathbf{s})}\left[\frac{\sqrt{|A|}}{\sqrt{|D|}}\sqrt{D_{ECQL}(\pi^*, \bar{\pi}_\beta)(\mathbf{s}) + 1}\right]$$
$$+ \alpha \frac{1}{1-\gamma}\mathbb{E}_{\mathbf{s} \sim d_{\hat{M}}^{\pi^*}(\mathbf{s})}[D_{ECQL}(\pi^*, \bar{\pi}_\beta)(\mathbf{s})]. \tag{16}$$

Note that all symbols here are in their non-evidential definitions which follow the same constraint on conservatism hyper-parameter $\alpha$ defined in (14). The explicitly expressed evidential term is contained in $D_{ECQL}(\pi^*, \bar{\pi}_\beta)(\mathbf{s})$. The proof for this statement is divided into two parts. The first part involves relating the return of $\pi^*$ in the empirical MDP $\hat{M}$ with the return of $\bar{\pi}_\beta$ in $\hat{M}$. Since, $\pi^*(\mathbf{a}|\mathbf{s})$ optimizes the penalized RL objective, it is the best policy under empirical MDP $\hat{M}$, and is guaranteed to behave better than the behavior policy $\bar{\pi}_\beta$ in a lowest bound governed by $\alpha \frac{1}{1-\gamma}\mathbb{E}_{\mathbf{s} \sim d_{\hat{M}}^{\pi^*}(\mathbf{s})}[D_{ECQL}(\pi^*, \bar{\pi}_\beta)(\mathbf{s})]$ as shown in the last term in Equation (16). The next step involves using concentration inequalities to upper and lower bound $J(\pi^*, \hat{M})$ and $J(\pi^*, M)$ and the corresponding difference for the behavior policy. According to Kumar et al.(Kumar et al., 2020), they apply Lemma D.4.1 to control such difference with $\gamma$, $R_{max}$, and $C_{T,\delta}, C_{r,\delta}$.

## C.3 PROOF OF THEOREM 2

We introduce the detailed derivative of Theorem 2 as given below:

$$J_{Q_c^e}(\theta) = \min_Q \alpha \left( \mathbb{E}_{\mathbf{s}\sim D, \mathbf{a}\sim\pi(\mathbf{a}|\mathbf{s})}[Q(\mathbf{a},\mathbf{s})] - \mathbb{E}_{\mathbf{s}\sim D, \mathbf{a}\sim\pi_\beta(\cdot|\mathbf{s})}[Q(\mathbf{a},\mathbf{s})] \right) + \frac{1}{2}\mathbb{E}_{\mathbf{s},\mathbf{a},\mathbf{s}'\sim D}\left[ \left( Q(\mathbf{a},\mathbf{s}) - \hat{\mathcal{B}}^\pi \hat{Q}^k(\mathbf{a},\mathbf{s}) \right)^2 \right]$$

$$= \min_Q \alpha \left( \mathbb{E}_{\mathbf{s}\sim D, \mathbf{a}\sim\pi(\mathbf{a}|\mathbf{s})}[Q(\mathbf{a},\mathbf{s})] - \mathbb{E}_{\mathbf{s}\sim D, \mathbf{a}\sim\pi_\beta(\cdot|\mathbf{s})}[Q(\mathbf{a},\mathbf{s})] \right)$$

$$+ \frac{1}{2}\mathbb{E}_{\mathbf{s},\mathbf{a},r_\pi^e,\mathbf{s}'\sim D,\mathbf{a}^*\sim\pi(\cdot|\mathbf{s}')}\left[ \left( Q(\mathbf{a},\mathbf{s}) - \left( r_\pi^e(\mathbf{s},\mathbf{a}) + \gamma \times \min\{\tilde{Q}^k(\mathbf{s}',\mathbf{a}^*), \hat{Q}^k(\mathbf{s}',\mathbf{a}^*)\} \right) \right)^2 \right]$$

$$= \min_Q \alpha \left( \mathbb{E}_{\mathbf{s}\sim D, \mathbf{a}\sim\pi(\mathbf{a}|\mathbf{s})}[Q(\mathbf{a},\mathbf{s})] - \mathbb{E}_{\mathbf{s}\sim D, \mathbf{a}\sim\pi_\beta(\cdot|\mathbf{s})}[Q(\mathbf{a},\mathbf{s})] \right) + \frac{1}{2}\mathbb{E}_{\mathbf{s},\mathbf{a},r_\pi^e,\mathbf{s}'\sim D,\mathbf{a}^*\sim\pi(\cdot|\mathbf{s}')}$$

$$\left( Q(\mathbf{a},\mathbf{s}) - \left( \underbrace{\frac{1}{N}\sum_{i\in\mathcal{N}_u}\text{rating}_{u,i}}_{\mathbf{r}} + \lambda\underbrace{\frac{1}{N}\sum_{i\in\mathcal{N}_u}\mathcal{U}_\pi(\cdot|\mathbf{e}_i)}_{\mathbb{R}} + \gamma \times \min\{\tilde{Q}^k(\mathbf{s}',\mathbf{a}^*), \hat{Q}^k(\mathbf{s}',\mathbf{a}^*)\} \right) \right)^2$$

$$= \min_Q \alpha \left( \mathbb{E}_{\mathbf{s}\sim D, \mathbf{a}\sim\pi(\mathbf{a}|\mathbf{s})}[Q(\mathbf{a},\mathbf{s})] - \mathbb{E}_{\mathbf{s}\sim D, \mathbf{a}\sim\pi_\beta(\cdot|\mathbf{s})}[Q(\mathbf{a},\mathbf{s})] \right)$$

$$+ \frac{1}{2}\mathbb{E}_{\mathbf{s},\mathbf{a},r_\pi^e,\mathbf{s}'\sim D,\mathbf{a}^*\sim\pi(\cdot|\mathbf{s}')}\left[ \left( \left( Q(\mathbf{a},\mathbf{s}) - \mathbf{r} - \gamma \times \min\{\tilde{Q}^k(\mathbf{s}',\mathbf{a}^*), \hat{Q}^k(\mathbf{s}',\mathbf{a}^*)\} \right) - \mathbb{R} \right)^2 \right].$$

We use $\hat{\mathcal{B}}^\pi \hat{Q}^k(\mathbf{a},\mathbf{s})$ to replace $\mathbf{r} + \gamma \times \min\{\tilde{Q}^k(\mathbf{s}',\mathbf{a}^*), \hat{Q}^k(\mathbf{s}',\mathbf{a}^*)\}$ and by some mathematical reshaping, we get:

$$J_{Q_c^e}(\theta) = \min_{\hat{Q}^k} \quad \frac{1}{2}\mathbb{E}_{\mathbf{s},\mathbf{a},r,\mathbf{s}'\sim D}\left[ \left( \hat{Q}^k(\mathbf{a},\mathbf{s}) - \hat{\mathcal{B}}^\pi \hat{Q}^k(\mathbf{a},\mathbf{s}) \right)^2 \right] + \alpha\mathbb{E}_{\mathbf{s}\sim D,\mathbf{a}\sim\bar{\pi}(\cdot|\mathbf{s})}[\hat{Q}^k(\mathbf{a},\mathbf{s})]$$

$$- \alpha \left( \mathbb{E}_{\mathbf{s}\sim D,\mathbf{a}\sim\bar{\pi}_\beta(\cdot|\mathbf{s})}(1+\mathbb{R})[\hat{Q}^k(\mathbf{a},\mathbf{s})] \right) + \underbrace{(\mathbb{R}^2 + \mathbb{R}\cdot\hat{\mathcal{B}}^\pi\hat{Q}^k(\mathbf{a},\mathbf{s}))}_{\mathcal{C}}. \quad (17)$$

where $\mathcal{C}$ is a constant not relating to the Q-value iterate.

## D EVIDENTIAL CONSERVATIVE Q-LEARNING ALGORITHM

We provided detail training procedure for the ECQL method in Algorithm 1.

## E DETAILED EXPERIMENTAL SETUP AND ADDITIONAL RESULTS

In this section, we provide additional details on the experiments, including datasets, setting, evaluation metrics, and comparison baselines. We also report more results, including comparison with additional baselines, an ablation study and a qualitative analysis.

### E.1 DETAILED EXPERIMENTAL SETUP

**Description of datasets.** We evaluate on four benchmark datasets that contain explicit ratings:

- **Movielens-1M**[1]: This dataset includes 1M explicit feedback (*i.e.,* ratings) made by 6,040 anonymous users on 3,900 distinct movies from 04/2000 to 02/2003.
- **Movielens-100K**[2]: This dataset contains 100,000 explicit ratings on a scale of (1-5) from 943 users on 1,682 movies. Each user at least rated 20 movies from September 19, 1997 through April 22, 1998.
- **Netflix** (Bennett et al., 2007): This dataset has around 100 million interactions, 480,000 users, and nearly 18,000 movies rated between 1998 to 2005. We pre-processed the dataset and selected 6,042 users with user-item interactions from 01/2002 to 12/2005.

---

[1]https://grouplens.org/datasets/movielens/1M/
[2]https://grouplens.org/datasets/movielens/100k/

---

**Algorithm 1** Evidential Conservative Q-Learning

---

**Require:** Hyperparameters: $\alpha, \lambda, \tau$, episode size $M$ and step size $T$

1: Initialize SSE: $\omega$, action network: $\phi$, evidence network: $\psi$, and critic network: $\theta$ , initial state: $\mathbf{s}_0$ and initial user embedding: $\mathbf{u}_0$ with $W^l = 2N$ items in sliding window $W_0$ from interaction history $H_u$, and Item Pool $\mathcal{I}$

2: **for** each epoch **do**

3:    **for** each user as an episode **do**

4:        **for** $t \in T$ **do**

5:            Compute state: $\mathbf{s}_t$ with (4).

6:            Compute action: $\mathbf{a}_t \sim \pi_\theta(.|\mathbf{s}_t)$

7:            Compute evidential score using (2)

8:            Recommend $top\text{-}N$ items based on computed evidential score to form a recommendation list $\mathcal{N}_u$.

9:            Compute rewards based on recommendation list $\mathcal{N}_u$ utilizing (3).

10:           Add $(\mathbf{s}_t, \mathbf{a}_t, r_\pi^e(\mathbf{s}_t, \mathbf{a}_t), \mathbf{s}_{t+1}, done)$ into replay buffer

11:           Move sliding window and take $\frac{W^l}{2}$ newly interacted items from $H_u$ and replace the other $\frac{W^l}{2}$ items with the top-N items from the recommendation list $\mathcal{N}_u$.

12:       **end for**

13:       **if** episode index number reaches $M$ **then**

14:           Sample batched data from replay buffer and forward into the networks.

15:           **while** not converged **do**

16:               Update critic network with (6)

17:               Update action network with (7)

18:               Update SSE network with (5)

19:               Update evidence network with (8)

20:           **end while**

21:       **end if**

22:   **end for**

23: **end for**

---

- **Yahoo! Music rating** (Dror et al., 2012): The dataset includes approximately 300,000 user-supplied ratings, and exactly 54,000 ratings for randomly selected songs. The ratings for randomly selected songs were collected between August 22, 2006 and September 7, 2006.
- **Movielens-10M** [3]: This data set contains 10,000,054 ratings applied to 10,681 movies by 71,567 users and released in January 2009. All users selected had rated at least 20 movies.
- **Amazon Book** Wang et al. (2019): This data set contains 2,984,108 ratings applied to 91,599 books by 52,643 users with at least ten interactions in each user sequence.

### E.1.1 EVALUATION METRICS

We use two standard metrics: Precision@N and nDCG@N, to measure the recommendation performance in all test users. We also use cumulative rewards as a measure of the average rating of recommendation for the RL-based methods comparison. For any particular user, we use a positive count directly to measure the performance.

- **Precision@N**: It is the fraction of the top-$N$ items recommended in each step of the episode that are positive (*i.e.,* rating $> \tau$) to the user. We average overall test users as the final precision.
- **nDCG@N**: Normalized Discounted Cumulative Gain (nDCG@N) measures ranking quality, considering the top-$N$ recommended items' information gain normalized by top-$N$ items of the ideal ranking list based on the ordered ground truth rating in each step of the RL episode. We average overall test users as the final precision.
- **Positive Count**: It is the total count of the top-$N$ items recommended in each step of the episode that are positive (*i.e.,* rating $> \tau$) to a particular test user.
- **Genre Count**: It is the total genre count of the top-$N$ items recommended in each step of the episode to a particular test user.
- **Cumulative Reward**: We evaluate test rewards based on the average ground-truth ratings of top-$N$ recommended items in each step within an RL episode. Cumulative Reward considers the sum of the test rewards before any step within an RL episode and averages across all test users.

### E.1.2 EXPERIMENTAL SETTING

We consider each user an episode for the RL setting and split users into 70% as training users and 30% as test users. For each user, we select the first $W^l = 10$ interacted items from history $H_u$ to represent an initial state $\mathbf{s}_0$. In the next state, we utilize previous state representation and concatenate with five item embeddings from the sliding window and the other five item embeddings from RL based recommendation to generate current state $\mathbf{s}_t$ by passing through the SSE module. Then, the action network generates mean and covariance for a Gaussian policy from which an action is sampled. This action is further passed to the evidence network, which utilizes the item embeddings of all user interacted items (Item Pool $\mathcal{I}$) to produce corresponding evidence for each item. We use the setting of classification, where explicit ground-truth ratings are used as class labels. With that evidence, we compute the evidential score by evaluating evidence-based rating and uncertainty to rank those items, which provides a list of top-$N$ ($N = 5$) final recommendations. We then evaluate the evidential reward based on their ground-truth ratings and information gain, functioning as the trade-off between exploitation and exploration. We set discounted factor $\gamma = 1$ and set $\tau = 3$ as a threshold to identify if an item is positive, *i.e.,* whether its ground-truth rating is larger than or equal to the threshold ($\text{rating}_{u,i} \geq \tau$). We implement and conduct our experiments based on PyTorch framework, with two A-100 GPUs, our model's parameter size is 25.6M and the inference speed of the whole RL system reaches an average 5 FLOPS.

### E.1.3 COMPARISON BASELINES

We compare with dynamic, sequential, and reinforcement learning models:

- **Dynamic models** include standard dynamic matrix factorization model timeSVD++ (Koren, 2009) as the time-evolving latent factorization model and collaborative Kalman filtering (CKF) (Gultekin & Paisley, 2014).
- **Sequential models** include Sequential Recommendation via Convolutional Sequence Embedding (Caser) (Tang & Wang, 2018), attention-based sequential recommendation model (SASRec) (Kang

---

[3] https://grouplens.org/datasets/movielens/10m/

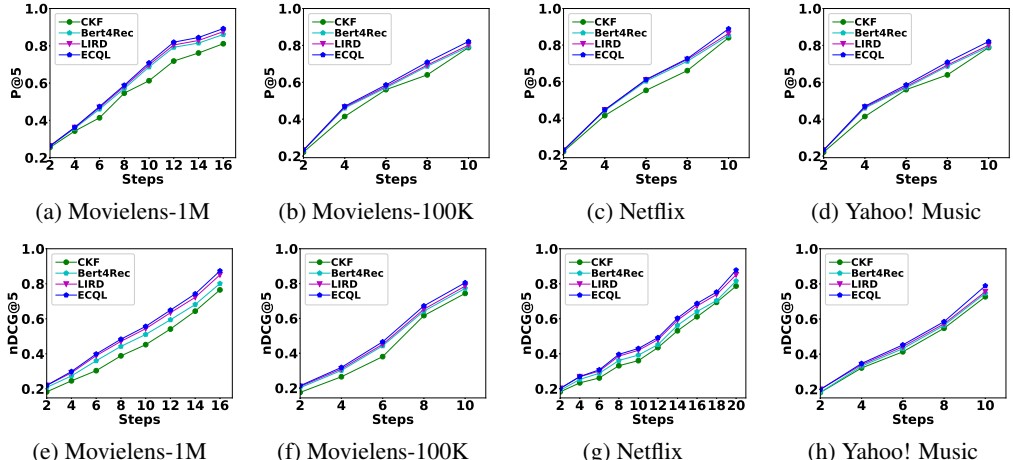

Figure 5: Performance comparison in each time step: (a)-(d): P@5; (e)-(h): nDCG@5

Table 5: Performance comparison (average P@N, nDCG@N across all test users)

| Model | MovieLens-1M | | | | |
|---|---|---|---|---|---|
| | P@5 | nDCG@5 | P@20 | nDCG@20 | Inference Time@20 (ms) |
| LIRD | 0.6137 | 0.5222 | 0.6238 | 0.5325 | 1.87 |
| CoLin | 0.6247 | 0.5285 | 0.6328 | 0.5372 | 0.88 |
| **ECQL** | **0.6379** | **0.5386** | **0.6471** | **0.5478** | **0.85** |

& McAuley, 2018), and sequential recommendation with bidirectional encoder (BERT4Rec) (Sun et al., 2019). For latest baselines, we have self supervised $S^3$-Rec (Zhou et al., 2020), contrastive learning based CL4SRec (Xie et al., 2022), and Seq2Seq (Ma et al., 2020) which reconstructs the representation of the future sequence as a whole and disentangle the intentions behind the given sequence of behavior.

- **Reinforcement learning-based models** include $\epsilon$-greedy (Zhao et al., 2013), deep Q-network based news recommendation (DRN) (Zheng et al., 2018), and actor-critic based list-wise recommendation (LIRD) (Zhao et al., 2017), and contextual bandit based method CoLin (Wu et al., 2016).

## E.2 ADDITIONAL RESULTS

In this section, we present additional experiments and compare with different types of baselines.

**Step-wise recommendation performance.** We further show the step-wise performance of both precision@5 (P@5) and nDCG@5 metrics considering top-5 recommended items in all datasets as shown in Figure 5. We show the average precision and nDCG of all the test users over each step after the model is fully trained to demonstrate the effectiveness of the dynamic recommendation. We fixed the step size to 16, 10, 20, and 10 for the Movielens-1M, Movielens-100K, Netflix, and Yahoo! Music datasets based on their average number of user-item interactions, respectively. At the initial steps, both precision and nDCG are low for all models (we choose the best model from each category as shown in Table 2). This is as expected due to lack of user interacted item observations. All the models start to improve after the initial stage. Dynamic models and sequential models still have poor performance compared to the RL-based methods. The proposed ECQL model provides consistently better performance over the entire process. However, it has a smaller advantage at the beginning due to its strong focus on exploration. After several step's observation, it quickly grasp user's interests and outperforms all its competitors by a clear margin. To further confirm this conclusion in a more realistic setting, we conduct additional experiments by comparing two strongest RL-based models using P@5, P@20, nDCG@5 and nDCG@20 metrics with our model. We also choose a subset of users from Movielens-1M with longer interaction sequences (> 180 interacted items). The experimental results are summarized in the Table 5. From the results, we can see that the proposed ECQL still demonstrates a clear advantage when recommending more items per time

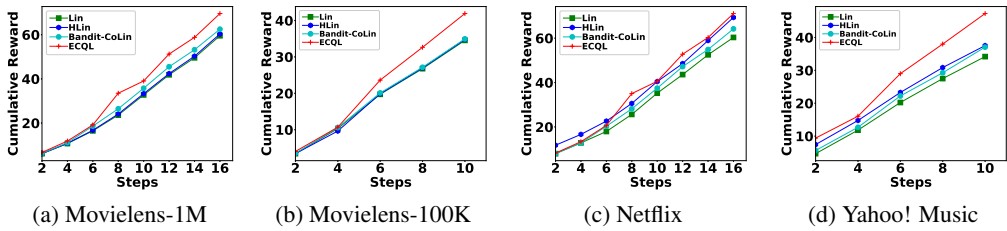

Figure 6: Comparison between bandit-based models and ours in four different datasets.

step, which further justifies ECQL's capability in capturing the user's long-term interests. Meanwhile, ECQL achieves high inference speed compared to its RL competitors, as indicated by the **Inference Time@20** column in Table 5.

**Comparison with contextual bandit based methods.** In the main paper, we compared with state-of-the-art RL based recommendation methods, including $\epsilon$-greedy (Zhao et al., 2013), deep Q-network based news recommendation (DRN) (Zheng et al., 2018), actor-critic based list-wise recommendation (LIRD) (Zhao et al., 2017), and collaborative contextual bandit based method CoLin (Wu et al., 2016). Here, we include two additional contextual bandit based models [4]: LinUCB (Lin) and Hybrid-LinUCB (HLin), to show a more complete bandit methods comparison in Figure 6. We report the cumulative reward in all four different datasets. ECQL shows a clear advantage over all these bandit based methods, which further justifies its better exploration capability to capture users' long-term interests.

**Comparison with other baselines.** In this section, we include two standard dynamic baselines: timeSVD++ and CKF and two recent models: DeepFM (Guo et al., 2017) and DCNv2 (Wang et al., 2021)for comparison. DeepFM integrates traditional factorization machine and deep learning to learn low- and high-order feature interactions. Similarly, DCNv2 is more expressive to learn feature interactions and cost-efficient at the same time. We also include one classical RL-based method called REINFORCE (Chen et al., 2019a), which applies off-policy learning to handle data bias. The test performance metric P@5 and nDCG@5 averaged over all test users across different time steps among the proposed ECQL and above three baselines in two datasets Movielens-1M and Movielens-100K are shown in Table 6 and Figure 7. Although these two deep learning-based recommender models achieve reasonable recommendation performance, they mainly lack to handle temporal preference of the users and hence perform worse than the proposed ECQL method. Furthermore, REINFORCE has limited exploration power and cannot effectively capture long-term user preference in the future, hence its performance is also lower than ECQL.

**Statistical testing results.** We have conducted statistical tests by running our model along with some competitive baselines three times and collecting the corresponding mean and standard deviation of P@5 and nDCG@5 performance on MovieLens-1M and MovieLens-100K datasets. The results are included in Table 7. It can be seen that the proposed model is more effective in performing effective exploration and provides more accurate recommendations, resulting in higher P@5 and nDCG@5 on both datasets considering the mean and variance in multiple runs. We further conduct a significance test to compare ECQL with the second best performing baseline CL4SRec. We obtain a p-value of 0.04, which confirms the performance advantage of ECQL over CL4SRec is statistically significant.

Table 6: Comparison of Recommendation Performance (average P@N and nDCG@N)

| Model | MovieLens-1M | | MovieLens-100K | |
|---|---|---|---|---|
| | P@5 | nDCG@5 | P@5 | nDCG@5 |
| timeSVD++ | 0.5341 | 0.4328 | 0.5034 | 0.4145 |
| CKF | 0.5567 | 0.4481 | 0.5285 | 0.4322 |
| DeepFM | 0.5647 | 0.4625 | 0.5428 | 0.4514 |
| REINFORCE | 0.6074 | 0.5116 | 0.5926 | 0.5149 |
| DCNv2 | 0.6152 | 0.5187 | 0.6158 | 0.5166 |
| **ECQL** | **0.6313** | **0.8735** | **0.6379** | **0.5386** |

---

[4] https://github.com/HCDM/BanditLib

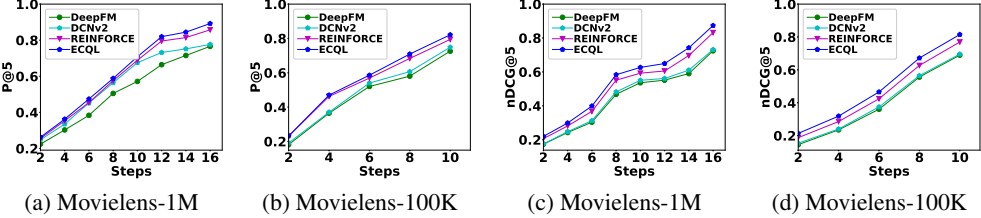

Figure 7: Step-wise performance comparison with other baselines. (a)-(b) P@5 (c)-(d) nDCG@5

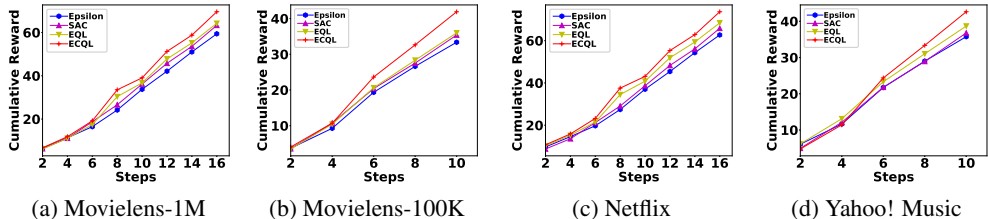

Figure 8: Comparisons among exploration strategies in four different datasets.

Table 7: Statistical Testing Results

| Model | MovieLens-1M | | MovieLens-100K | |
|---|---|---|---|---|
| | P@5 | nDCG@5 | P@5 | nDCG@5 |
| Seq2Seq | 0.5818±0.025 | 0.4752±0.018 | 0.5579±0.023 | 0.4614±0.018 |
| $S^3$-Rec | 0.6108±0.022 | 0.4926±0.016 | 0.5792±0.021 | 0.4767±0.016 |
| CL4SRec | 0.6135±0.019 | 0.4952±0.014 | 0.5813±0.018 | 0.4781±0.012 |
| **ECQL** | **0.6313± 0.028** | **0.5365± 0.021** | **0.6379± 0.022** | **0.5386± 0.016** |

**Comparison with RL-based sequential methods on larger datasets.** To experiment SAR, we allow the sequence adaptation action network to provide sequence length up to the fixed sequence length, *i.e.,*, 10 in each sequence to match our setup. Similarly for ResAct, we use the 5 rating scale in both datasets to provide long-term engagement.

Table 8: Comparison with recent RL-based sequential baselines in large datasets

| Model | ML-10M | | Amazon Book | |
|---|---|---|---|---|
| | P@5 | nDCG@5 | P@5 | nDCG@5 |
| SAR | 0.5374 | 0.4762 | 0.4915 | 0.3327 |
| ResAct | 0.5624 | 0.5136 | 0.5273 | 0.3654 |
| **ECQL** | **0.6425** | **0.5518** | **0.6017** | **0.4896** |

From Table 8 above, it is clear that our method is taking advantage of systematic exploration and providing better performance in both larger datasets. In comparison to SAR, the ResAct has better performance due to its capability to leverage a residual network to strengthen its user representation capability. However, it also fails to perform enough exploration and achieves poor performance as compared with our method.

E.3  ABLATION STUDY

We further investigate the effect of exploration in our proposed model by comparing ECQL with an alternative design without vacuity (denoted as Epsilon) or conservative learning (denoted as EQL) guided exploration. Furthermore, we also compare exploration using the first-order uncertainty, which is employed by soft-actor-critic (SAC). We visualize the comparison results on four datasets in Figure 8. It shows that without uncertainty-guided exploration, the model collects the least cumulative reward in the long run. SAC utilizes entropy-based exploration and achieves better cumulative reward

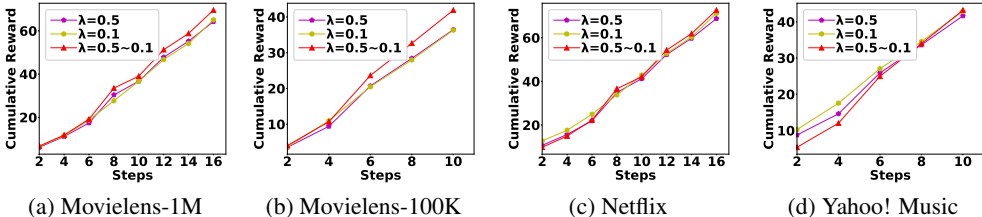

(a) Movielens-1M     (b) Movielens-100K     (c) Netflix     (d) Yahoo! Music

Figure 9: Average cumulative reward of ECQL for different $\lambda$

than without uncertainty-guided exploration. This provides evidence that the role of the exploration is crucial in RL-based recommendations. As shown in Figure 8, its performance is worse than the vacuity-based ECQL method. This is because vacuity-guided exploration allows our model to focus its exploration on the most informative items that help the model gain the most knowledge to form an optimal policy. However, given the sparse reward space in RS field, we need to guarantee that the recommendation is safe and the exploration is guided by a conservative learning that prefers the stable behavior policy. As shown in Figure 8, EQL performance is worse than ECQL by a clear margin, especially in the later steps when the model gets richer knowledge about the user through interaction. The advantage of ECQL over entropy-based exploration or exploration without a conservative constraint is also consistent with our earlier discussion in Section 4.4 in the main paper.

**Impact of hyperparameter ($\lambda$).** The hyperparameter ($\lambda$) plays a critical role in recommending the top-$N$ items and generating the evidential reward. We test three different settings: $\lambda = 0.1$, $\lambda = 0.5$, and gradually reducing $\lambda$ from $0.5$ to $0.1$. As can be seen from Figure 9, dynamically adjusting $\lambda$ achieves consistently better performance on all datasets. This supports the intuition that in the early steps, a large $\lambda$ allows the model to conduct sufficient exploration. Once the model gains sufficient knowledge from the environment and is able to make accurate recommendations, reducing $\lambda$ will allow the model to exploit its knowledge to provide effective recommendations.

**Impact of each module.** We evaluate the impact of each module by training the model with "RNN", "CEAC" and "RNN+CEAC" configurations as detailed below:

- "RNN" means we use RNN as our sequential encoder and use an MLP based classifier to predict the rating for each item in the item pool for both training and test user sets.

- "CEAC" means that we only leverage current user embedding to be the action network's input and continue the RL process without any sequential state encoder to help maintain the state embedding.

- "RNN+CEAC" means that two modules work together as our ECQL framework.

We calculate the P@5 and nDCG@5 for each setting in the Movielens-1M test set. The results show that the integration of the two modules significantly outperform each individual module.

Table 9: Impact of each module

| Modules | P@5 | nDCG@5 |
|---|---|---|
| RNN | 0.4899 | 0.4175 |
| CEAC | 0.6044 | 0.5108 |
| RNN + CEAC (i.e. ECQL) | 0.6313 | 0.5365 |

**Impact of sequence encoders:** In this section, we provide the results of different sequence encoders, including RNN, LSTM, and GRU in Movielens-1M test set. The result shows that our model's performance remains robust to the type of the sequence encoder.

Table 10: Impact of sequential encoders

| Models | P@5 | nDCG@5 |
|---|---|---|
| RNN + CEAC | 0.6313 | 0.5365 |
| LSTM + CEAC | 0.6312 | 0.5365 |
| GRU + CEAC | 0.6315 | 0.5366 |

**Impact of exploration strategies:** In this section, we compare our ECQL with methods leveraging different exploration strategies and report the comparison performance in Movielens-1M test set. Specifically,

- $\epsilon$-greedy (AC, QL) is a basic exploration strategy, which uses actor-critic to conduct Q-learning. Since we only set hyper-parameter $\epsilon = 0.1$, which is our best practice after grid-search, there's no need to combine conservative learning in this exploration direction.
- SAC also leverages actor-critic to conduct Q-learning. The difference comparing to $\epsilon$-greedy is that it uses first-order uncertainty "Entropy" in the rating prediction for score-based exploration and the reward design.
- EQL is our framework with traditional Q-learning in stead of conservative Q-learning, so the exploration is purely guided by uncertainty without any constraints.
- CoLin is a collaborative contextual bandit based method which explicitly models the underlying dependency among users/bandits with the help of a weighted adjacency graph, comparing to other bandit exploration technique like upper confidence bound based (UCB-based) HLin or factorUCB, it achieves best performance in multiple recommendation datasets.

Table 11: Impact of Exploration Strategies

| Models | P@5 | nDCG@5 |
|---|---|---|
| $\epsilon$-greedy (AC, QL) | 0.5977 | 0.4834 |
| SAC | 0.6105 | 0.5215 |
| EQL | 0.6234 | 0.5331 |
| CoLin | 0.6162 | 0.5216 |
| ECQL | 0.6313 | 0.5365 |

### E.4 QUALITATIVE ANALYSIS

We conduct an additional qualitative analysis of a particular user in the Netflix dataset by comparing the top-$N$ recommended items in time step 16 of our proposed method ECQL with other models using different exploration strategies, *e.g.,* $\epsilon$-greedy, SAC, and CoLin. We also include EQL to further validate the necessity of conservative Q-learning. As shown in Table 12, a user with ID 254775 has five most frequently watched movie categories. By order, they are **Action**, **Sci-Fi**, **Adventure**, **Drama**, and **Romance**, which form a top-5 frequency list and are bold in the table. From the table, we observe that ECQL successfully captures user's long-term interests by recommending important movies that all belong to the top-5 frequency list. Compared to ECQL, EQL recommends some irrelevant movies of Documentary and Educational Genres that deteriorate the performance since it is not constrained by a conservative view in the Q-learning process. Frequently recommending totally irrelevant movies may pose a higher risk of losing the user. For different exploration strategies, $\epsilon$-greedy brings the least performance by recommending only two important movies that lie in the top-5 frequency list as it does not leverage a systematic exploration way to explore the user's potential interests. In comparison, bandit-based CoLin and entropy-based SAC leverage bandit theory and first-order entropy to measure the uncertainty and both lead to an improved recommendation performance by recommending at least three important high frequent movies out of five that reflect the user's interests. However, we emphasize that they only capture the short-time interest of the user, which is demonstrated by their recommended important items' relative order in the top-5 frequency list.

The above analysis is from a random test user in the Netflix data set. For a more comprehensive analysis, we calculate the Gini index of the diversity for all the recommended results. First, we

Table 12: Recommended movies for UserID: 254775

| Model | Recommended Important Movies | Movie Genre | Vacuity |
|---|---|---|---|
| | 1995,Star Trek: Voyager: Season 1 | **Sci-Fi,Action** | 0.10 |
| | 2005,7 Seconds | **Action,Adventure** | 0.12 |
| ECQL | 1994,Immortal Beloved | **Romance** | 0.14 |
| | 1996,No Way Back | **Action** | 0.11 |
| | 1996,Screamers | **Sci-fi, Action** | 0.09 |
| | 2003,Dinosaur Planet | **Adventure**,Fantasy | 0.16 |
| | 1982,Nature: Antarctica | Documentary | 0.15 |
| EQL | 1997,Sick | **Romance, Drama** | 0.14 |
| | 1994,Paula Abdul's Get Up | Educational | 0.16 |
| | 1997,Character | **Romance,Drama** | 0.15 |
| | 1979,Winter Kills | **Action** | 0.08 |
| | 1991,Antarctica: IMAX | Documentary | 0.16 |
| $\epsilon$-greedy | 1951,The Frogmen | **Drama** | 0.13 |
| | 1983,Silkwood | Documentary | 0.15 |
| | 2002,The Powerpuff Girls Movie | Animation | 0.15 |
| | 1951,The Lemon Drop Kid | Comedy | 0.13 |
| | 2002,Mostly Martha | **Romance** | 0.13 |
| SAC | 1989,A Fishy Story | **Romance** | 0.14 |
| | 2004,Spartan | **Action**,Military | 0.11 |
| | 1997,The Game | **Action** | 0.09 |
| | 1965,The Great Race | **Adventure,Action** | N/A |
| | 2002,Obsessed | **Romance,Drama** | N/A |
| CoLin | 2000,Magnolia: Bonus Material | Military | N/A |
| | 1965,The Battle of Algiers: Bonus Material | Military | N/A |
| | 1972,Seeta Aur Geeta | Musical,**Romance** | N/A |

test 1,800 users from Netflix test set and then collect 16 (number of time steps) × 1,800 total recommendations. For each recommendation, we calculate its Gini index by separating 5 results into different categories and then calculate

$$Gini = 1 - \sum_{c=1}^{C} P(c)^2 \tag{18}$$

where $C$ is the number of categories and $P(c), c \in [1, C]$ is the probability for each category. Finally, we average the calculated Gini index for 1,800 recommendations and get the averaged Gini index as 0.71, which is close to 1. This indicates that the recommendation of our model is quite diverse and contains objects from different categories. Comparing to SAC and CoLin which have averaged Gini indexes 0.65 and 0.68 respectively, our ECQL achieves the most diverse recommendation thanks to the novel evidence-based exploration, which is also verified by Table 12.

## F  BROADER IMPACT, LIMITATIONS, AND FUTURE WORK

In this section, we first describe the potential broader impacts of our work. We then discuss the limitations and identify some possible future directions.

### F.1  BROADER IMPACT

The proposed ECQL can be generally applied to safety-critical applications, which are common in many domains such as health, autonomous vehicle, cybersecurity, military operations, and more. We provide conservative evidential exploration which doesn't let learned policy deviate far from the behavioral policy and supports the gradual exploration of a complex environment with sparse reward signals by leveraging fine-grained second-order uncertainties. The principled exploration can ensure high information gain with much reduced data annotations, which can benefit many domains where data annotation is costly.

### F.2  LIMITATIONS AND FUTURE WORKS

In our evaluation, we applied the proposed ECQL method to the simulated online setting using the offline user interaction datasets to mimic real-world scenarios. It could be more interesting to implement this ECQL method in the online datasets which could better reflect the model's real-time performance. We are planning to further extend this work to apply to those real-world time sequence

data in different fields such as health and autonomous vehicle to see its scalability, generalizability, and adaptability as a future work. And we will also conduct plenty of ablation and case analysis in these sparse and safety-critical domains to evaluate the effectiveness of our unique evidence-based exploration balanced with a conservative learning design.

## G   SOURCE CODE

The source code and processed datasets can be accessed here. `https://anonymous.4open.science/r/EvidentialRecommendation-2BDE/README.md`

