# OpenReview forum: "Evidential Conservative Q-Learning for Dynamic Recommendations"
_ICLR.cc/2024/Conference — Submitted to ICLR 2024_

### Official Review · Reviewer_gFnt · 2023-10-30

**Soundness:** 3 good
**Presentation:** 2 fair
**Contribution:** 2 fair
**Rating:** 5
**Confidence:** 5

**Summary:**

The paper introduces the Evidential Conservative Q-Learning (ECQL) framework designed for dynamic recommendations. ECQL employs evidence-based exploration, guided by uncertainty measures, to uncover items that cater to the users' long-term preferences. The proposed framework is characterized by its Sequential state encoder, which keeps track of a dynamic state space via a sliding window over user history, and the Conservative Evidential Actor-Critic (CEAC) module that facilitates uncertainty-informed exploration and conservative policy learning. The framework's distinctive elements include an evidential reward system that leans towards exploring items with uncertain outcomes to maximize information gain, and a conservative Q-learning mechanism that guards against overestimating the policy value, thus ensuring quality recommendations. The paper further supports the efficacy of ECQL with a theoretical analysis that underscores its convergence behavior and recommendation quality. Empirical evaluations on real-world datasets reveal that ECQL surpasses benchmark methods. Additionally, ablation studies emphasize the significant contributions of vacuity-driven exploration and conservative learning. The major contributions of the paper encompass the innovation of an evidential RL approach for recommendations, an emphasis on uncertainty-driven exploration, the integration of conservative off-policy learning, theoretical assurances, and the creation of a holistic end-to-end framework.

**Strengths:**

1. The paper stands out due to its innovative combination of evidential learning and reinforcement learning tailored for recommendation systems. The inception of leveraging evidence-based uncertainty for exploration, paired with conservative off-policy learning to dodge overestimations, showcases an inventive approach.

2. On the technical front, the paper exhibits robustness. It provides theoretical backing, ensuring that the methods are grounded in solid logic. The empirical evaluations, especially those conducted on real-world datasets, further validate the methodology by demonstrating its state-of-the-art prowess.

3. Structurally, the paper is well-organized, offering a lucid exposition of the principal concepts and the constituents of the framework.

**Weaknesses:**

My main concern about the paper is the experimental parts.

1. The datasets used are very small in this paper. Ml-1M and ML-100K are both very small. For Netﬂix, the authors only sampled 6,042 users. For Yahoo!, only 54,000 ratings were sampled. Larger datasets are required for validation.

2. I'm concerned about the effect-boosting effect of the proposed method on different datasets. Since the improvment is not large. Did the authors perform the hypothesis testing to prove the significance of improvements? Also, the effect enhancement of the proposed method is much smaller inthe  largest dataset  Yahoo! than in other datasets. Is there any specific research on this phenomenon?

3. In the experimental part, the RL-based models lack some more recent work (the latest method is in 2018), which cuts down the validity of the proposed method.

4. Some diagrams can bedemonstrated in a more expressive way. For example, Figure 2 could insert some images to illustrate the composition of themodules in the network.

5. Some formulas are not written very rigorously, for example: all mathematical formulas are not followed by a punctuation mark. In addition, some expressions are not sufficiently formal, such as 'score' and 'rating'.

**Questions:**

In the QUALITATIVE STUDY in Section 5.2, regarding the random selection of the User ID, is there a discussion of the reason for selecting this user? I think it is more persuasive to describe the reason for the selection first and select multiple users for the QUALITATIVE STUDY.

---

> ### Author Response · Authors · 2023-11-19
> **Response to Reviewer gFnt**
>
> Thank you for reviewing our paper and providing valuable comments. We summarize our response as follows:
>
> **Q1. The datasets used are very small in this paper. Ml-1M and ML-100K are both very small. For Netﬂix, the authors only sampled 6,042 users. For Yahoo!, only 54,000 ratings were sampled. Larger datasets are required for validation.**
>
> Thanks for the suggestion. Please refer to the answer to Q2 in the general response.
>
> **Q2. I'm concerned about the effect-boosting effect of the proposed method on different datasets. Since the improvement is not large. Did the authors perform the hypothesis testing to prove the significance of improvements? Also, the effect enhancement of the proposed method is much smaller in the largest dataset Yahoo! than in other datasets. Is there any specific research on this phenomenon?**
>
> Thank you for this comment. We have provided the standard deviation of the proposed ECQL model along with the most competitive baselines from each category in Table 7 of Appendix E.2. The small standard deviation clearly the performance advantage over other methods. We further conduct a significance test to compare ECQL with the second best performing baseline CL4SRec. We obtain a p-value of 0.04, which confirms the performance advantage of ECQL over CL4SRec is statistically significant. To better understand the impact on the size of the datasets, we have conducted additional experiments on two larger datasets as suggested by the reviewer. As can be seen from the answer to Q2 in the general response, our model achieves much better P@5 and nDCG@5 as compared with two most recent RL based baselines.
>
> **Q3. In the experimental part, the RL-based models lack some more recent work (the latest method is in 2018), which cuts down the validity of the proposed method.**
>
> Thanks for the suggestion. Please refer to the answer to Q1 in the general response.
>
>
>
> **Q4. Some diagrams can be demonstrated in a more expressive way. For example, Figure 2 could insert some images to illustrate the composition of the modules in the network.**
>
> Thanks for the suggestion. We updated Figure 2 in the revised paper to better illustrate the modules composition.
>
> **Q5.  Some formulas are not written very rigorously, for example: all mathematical formulas are not followed by a punctuation mark. In addition, some expressions are not sufficiently formal, such as 'score' and 'rating'.**
>
> Following the reviewer's suggestion, we have added punctuation marks to all the equations. The mathematical definitions of score and rating are given in Equation 2 and the following text, respectively. We are happy to make them more formal if the reviewer has more specific suggestions.
>
>
> **Q6. Describe the reason for the selection of the given user and then select multiple users for the qualitative study.**
>
> Thanks for the comment. We would like to clarify that the users in Tables 1, 3, and 12 of the revised paper are all different users randomly selected from the test set. They are representatives from many testing users where we provide a more detailed analysis on the recommendation behavior of ECQL and the competitive baselines. It is clear that ECQL needs to perform better on those individual users in order to achieve an overall better recommendation performance on evaluation metrics, including precision and nDCG. Following the reviewer's suggestion, we conduct a more comprehensive analysis on multiple users to complement the qualitative study presented in Appendix E.4. We calculate the Gini index of the diversity for all the recommended results. First, we test 1,800 users from Netflix test set and then collect 16 (number of time steps) $\times$ 1,800 total recommendations. For each recommendation, we calculate its Gini index by separating 5 results into different categories and then calculate
> \begin{align}
> Gini=1-\sum_{c=1}^{C}P(c)^2
> \end{align}
> where $C$ is the number of categories and $P(c), c\in [1,C]$ is the probability for each category. Finally, we average the calculated Gini index for 1,800 recommendations and get the averaged Gini index as 0.71, which is close to 1. This indicates that the recommendation of our model is quite diverse and contains objects from different categories. Comparing to SAC and CoLin which have averaged Gini indexes 0.65 and 0.68 respectively, our ECQL achieves the most diverse recommendation thanks to the novel evidence-based exploration, which is also verified by Table 12 of the revised paper.

---

> > ### Author Response · Authors · 2023-11-22
> >
> > Dear Reviewer gFnt,
> >
> > Thanks for your questions and valuable suggestions, which help us to further improve the paper. We summarize our main responses as below:
> >
> > 1. We provide the standard deviation of the proposed ECQL model along with the most competitive baselines from each category to justify the performance advantage is statistically significant. We also conduct a significance test by comparing the second best baseline "CL4SRec" and report the (small) p-value.
> >
> > 2. We compare with latest RL-based sequential models SAR (Antaris et al., 2021) and ResAct (Xue et al., 2023) on two larger datasets MovieLens-10M and Amazon Book. We report the results in our answer to Q2 in the general response.
> >
> > 3. We clarify that the users in Tables 1, 3, and 12 of the revised paper are all different representative users randomly selected from the test set. Following the reviewer's suggestion, we conduct a more comprehensive analysis on multiple users to complement the qualitative study presented in Appendix E.4 and calculate the Gini index of the diversity for all the recommended results to give a quantitative evidence.
> >
> >
> > As always, we are happy to provide any additional clarifications if needed.

---

### Official Review · Reviewer_Yout · 2023-11-01

**Soundness:** 2 fair
**Presentation:** 2 fair
**Contribution:** 2 fair
**Rating:** 3
**Confidence:** 4

**Summary:**

This paper proposes a new variant of Q-learning called ECQL for dynamic recommendations problems. The ECQL aims to improve the exploration efficiency of \epsilon-greedy strategy, which is commonly used in many reinforcement learning algorithms. Specifically, the exploration strategy of ECQL integrates evidence-based uncertainty and conservative learning. The purpose is to discover items that are located beyond current observation but reflect users’ long-term interests. Extensive experiments are conducted to validate the performance of ECQL.

**Strengths:**

This paper studies an important problem of dynamic recommendation, i.e., discover items that are located beyond current observation but reflect users’ long-term interests.

The proposed ECQL is shown to have nice empirical performance.

**Weaknesses:**

The motivation of this work is not convincing. It is claimed that epsilon-greedy may not be able to learn the optimal policy that captures effective user preferences and achieves the maximum expected reward over the long term. This claim is not supported by any evidences. To the best of my knowledge, the \epsilon-greedy strategy is a simple approach to balance the exploration vs. exploitation tradeoff of RL. As long as the RL model is tailored to RS in a right manner and the exploration vs. exploitation tradeoff is well balanced, the learned policy may not have the claimed weakness. Also, one can tune epsilon to attain different strength of exploration. There are other exploration strategies developed in RL literature. Can you claim all of them are not learn the optimal policy that captures effective user preferences and achieves the maximum expected reward over the long term? I am also confused by Figure 1. Little details are provided. It is hard for me to judge whether Figure 1 is the outcome of improper applying of RL to RS or some other factors. For example, is it a consequence of that the optimal policy of the RL has such weakness or the epsilon-greedy strategy leads to a sub-optimal strategy that has such weakness.

The challenge analysis is confusing. It is mentioned in the Introduction “to address the above challenge”. I do not get what is the above challenge. Technically, what is it?

This paper is not placed clearly. It aims to address the limitation of epsilon-greedy strategy. In the related work, it does not discuss previous works on exploration strategies and place this paper properly in this research line. In the related work, I am not convinced by the claim that randomize exploration strategies are less effective at capturing users’ long-term preferences. Could provide any evidence?

It is not clear how the proposed exploration strategy connects to balancing exploration vs. exploitation tradeoff. In terms of the balancing exploration vs. exploitation tradeoff, it is unclear why it is better than the epsilon-greedy or fine tuned epsilon-greedy.

**Questions:**

Please refer to the comments on the weakness.

**Details Of Ethics Concerns:**

No.

---

> ### Author Response · Authors · 2023-11-19
> **Response to Reviewer Yout [Part I]**
>
> Thank you for reviewing our paper and providing valuable comments. We first summarize the proposed model (ECQL), its important concept, and its implication in RS. Second, we provide our responses to the questions.
>
> **Motivation and novelty of ECQL**: In general, ECQL learns an effective and cautious recommendation policy by integrating evidence-based uncertainty and conservative learning systematically. ECQL conducts evidence-aware explorations to discover items that are located beyond current observation but potentially aligned with users’ long-term interests. Effective exploration of ECQL is guaranteed through the novel use of vacuity, which is an evidence-based second-order uncertainty, derived in the subjective logic framework (see Equation 1 for the definition of vacuity).  The vacuity guided exploration can identify uncertain and informative items (from large item space), indicative of users' long-term interest. As a result, the proposed evidential reward (see Equation 3) encourages the RL agent to recommend items that the model has the least knowledge (as indicated by a high vacuity). After collecting the user feedback, the RL agent can most effectively gain knowledge of the user preference to make better recommendations in the long run. Furthermore, by integrating uncertainty-aware exploration with conservative learning, ECQL is able to make more diverse recommendations that may reflect a long-term interest while keeping a conservative view that does not deviate too much from current interests. Please refer to  Table 12 of the Appendix for more illustrative examples and supporting evidence. We also provide a comprehensive quantitative comparison among different exploration strategies in our answer to Q3 of the general response.
>
>
> **Q1. It is claimed that $\epsilon$-greedy may not be able to learn the optimal policy that captures effective user preferences and achieves the maximum expected reward over the long term. This claim is not supported by any evidence.**
>
> Since $\epsilon$-greedy performs completely random exploration, it is very likely that the explored items may contain the ones that the user is already familiar with or those that completely deviate from users' interest. In contrast, as mentioned above, the vacuity-guided exploration encourages the RL agent to recommend items that the model has the least knowledge. After collecting the user feedback, the RL agent can most effectively gain knowledge of the user preference to make better recommendations in the long run. Besides $\epsilon$-greedy, we also compared with another commonly used exploration strategy, soft-actor-critic (SAC), which leverages entropy to perform exploration. However, as we made clear in the paper, a high entropy may imply either high vacuity (lack of evidence) or high dissonance (conflict of strong evidence). However, dissonance is not effective for exploration in RS due to its focus on confusing items mostly derived based on the users' current interests. Figure 8 of the Appendix shows that ECQL achieves a much higher cumulative reward than SAC and $\epsilon$-greedy, which confirms its effectiveness in capturing users' long-term interest. Table 12 of Appendix E.4 presents a qualitative study to demonstrate that ECQL is able to recommend more diverse items than SAC and $\epsilon$-greedy relevant to user's preference. Finally, Table 11 of Appendix E.3 provides a comparison with a collaborative contextual bandit based method CoLin to further demonstrate the effective exploration of ECQL.
>
>
> **Q2.  For Figure 1. Little details are provided.  It is hard for me to judge whether Figure 1 is the outcome of improper applying of RL to RS or some other factors.**
>
> In Figure 1, we illustrate that the existing RL methods based on $\epsilon$-greedy do not provide effective exploration. Their recommendations usually concentrate on a narrower items space, primarily focusing on highly rated items but only reflecting short-term interest. In contrast, the proposed ECQL method provides more informative items to help models learn the user's long-term interest.
>
>
> **Q3. It is mentioned in the Introduction "to address the above challenge". I do not get what is the above challenge. Technically, what is it?**
>
> As mentioned in the paper and the response to Q2, the major challenge with the existing RL-based methods is the lack of a systematic exploration to discover users' long-term interests. Hence, we need a better exploration strategy to capture user's diverse interests and maximize the long-term reward.
>
>
> **Q4. I am not convinced by the claim that randomize exploration strategies are less effective at capturing users' long-term preferences. Could provide any evidence?**
>
> Please refer to our detailed response to Q1 that mentions multiple concrete evidences provided in the paper to explain why random exploration is less effective.

---

> ### Author Response · Authors · 2023-11-19
> **Response to Reviewer Yout [Part II]**
>
> **Q5.  In terms of the balancing exploration vs. exploitation tradeoff, it is unclear why it is better than the epsilon-greedy or fine-tuned $\epsilon$-greedy.**
>
> To achieve optimal policy in the guided exploration system, it is always important to balance exploration and exploitation. By leveraging second-order uncertainty (vacuity) guided exploration, we can effectively locate items that the model lacks the knowledge for now but may reflect the model's long-term interest to build a more precise user profile. Please refer to our discussion on the motivation and novelty of ECQL for more details.

---

> > ### Author Response · Authors · 2023-11-22
> >
> > Dear Reviewer Yout,
> >
> > Thank you for reviewing our paper and providing valuable comments. Specifically, we explain the major limitations of the existing RL-based methods and introduce the motivation and novelty of ECQL by elaborating that the systematic integration of evidence-based uncertainty and conservative learning is important to an effective and cautious recommendation policy. In contrast, $\epsilon$-greedy conducts random exploration with no control, so it is very likely that the explored items may contain the ones that the user is already familiar with or those that completely deviate from users' interest. Further, we compare with $\epsilon$-greedy and other RL exploration strategies such as soft-actor-critic (SAC) and a collaborative contextual bandit based method CoLin to demonstrate the superiority of vacuity-guided exploration.

---

### Official Review · Reviewer_GVwA · 2023-11-02

**Soundness:** 3 good
**Presentation:** 3 good
**Contribution:** 3 good
**Rating:** 5
**Confidence:** 4

**Summary:**

In this paper, the authors introduce a novel evidential conservative Q-learning framework (ECQL) that learns an effective and conservative recommendation policy by integrating evidence-based uncertainty and conservative learning. Specifically, ECQL includes two main components, i.e., a uniquely designed sequential state encoder and a novel conservative evidential-actor-critic module. Extensive experiments on real datasets demonstrate the effectiveness of the proposed method.

**Strengths:**

1. This paper introduces a novel recommendation framework named ECQL, which uses evidential conservative Q-learning for dynamic recommendation.

2. ECQL is a new model that integrates reinforcement learning with evidential learning to provide uncertainty-aware diverse recommendation.

3. The authors also provide theoretical analysis to justify the desired convergence behavior and recommendation quality that guarantees to avoid risky recommendations.

4. The proposed framework integrates a sequential encoder, an actor-critic network, and an evidence network to provide end-to-end integrated training process.

5. The authors have performed extensive experiments on real datasets and compare the proposed model with SOTA baseline methods.

**Weaknesses:**

1. In the Section 2, the authors do no include the recent studies about sequential recommendation. The most recent dynamic/sequential recommendation methods mentioned by them are proposed in 2019. The sequential recommendation methods developed in recent 4 year are not discussed in Section 2. Similarly, the authors are suggested to include some discussions about the RL methods developed in recent 3 years.

2. One advantage of the proposed method is to provide diverse recommendations. However, in Table 2, the authors do not analyze the diversity of the generated recommendation results.

3. The authors are suggested to include more sequential recommendation methods proposed in 2021 and 2022 as baselines. TimeSVD++ and CKF are not needed to be used as baseline methods.

4. The Movielens-100K dataset is too small for experimental evaluation. The authors are suggested to include some larger datasets for experiments, for example Amazon review datasets.

5. In the proposed ECQL, there are two main modules. However, there is no experiments studying the importance of these two parts. The authors are suggested to perform an ablation study.

**Questions:**

1. The proposed framework integrates a sequential encoder, an actor-critic network, and an evidence network to provide end-to-end integrated training process. For the actor-critic network and evidence network, which one is more important? Moreover, whether the model performance is dominated by the base sequential encoder? Whether the proposed framework can help improve the performance of different base sequential encoders?

---

> ### Author Response · Authors · 2023-11-19
> **Response to Reviewer GVwA**
>
> Thank you for reviewing our paper and providing valuable comments. We summarize our response as follows.
>
> **Q1. In the Section 2, the authors do not include the recent studies about sequential recommendation... Similarly, the authors are suggested to include some discussions about the RL methods developed in recent 3 years.**
>
> Thanks for the suggestion. Please refer to the answer to Q1 in the general response.
>
>
> **Q2.  One advantage of the proposed method is to provide diverse recommendations. However, in Table 2, the authors do not analyze the diversity of the generated recommendation results.**
>
> Thank you for the suggestion. We have provided the qualitative analysis of diversity in Table 12 of the Appendix. To further provide the quantitative evidence, we calculate the Gini index of the diversity for all the recommended results. First, we test 1,800 users from Netflix test set and then collect 16 (number of time steps) $\times$ 1,800 total recommendations. For each recommendation, we calculate its Gini index by separating 5 results into different categories and then calculate
> \begin{align}
> Gini=1-\sum_{c=1}^{C}P(c)^2
> \end{align}
> where $C$ is the number of categories and $P(c), c\in [1,C]$ is the probability for each category. Finally, we average the calculated Gini index for 1,800 recommendations and get the averaged Gini index as 0.71, which is close to 1. This indicates that the recommendation of our model is quite diverse and contains objects from different categories. Comparing to SAC and CoLin which have averaged Gini indexes 0.65 and 0.68 respectively, our ECQL achieves the most diverse recommendation thanks to the novel evidence-based exploration, which is also verified by Table 12 of the revised paper.
>
>
>
>
> **Q3. The authors are suggested to include more sequential recommendation methods proposed in 2021 and 2022 as baselines. TimeSVD++ and CKF are not needed to be used as baseline methods.**
>
> Thank you for the suggestion. We have removed TimeSVD++ and CKF from the main comparison table and moved them to Appendix E.2 where we compare with some other baseline models. For the newer sequential recommendation models, in addition to CL4SRec (Xie et al., 2022), we have added two additional baselines: ResAct (Xue et al., 2023) and SAR (Antaris et al., 2021), and compared them with our approach on two larger datasets. Please refer to our response to Q4 below for details.
>
>
> **Q4. The Movielens-100K dataset is too small for experimental evaluation. The authors are suggested to include some larger datasets for experiments, for example, Amazon review datasets.**
>
> Thanks for the suggestion. Please refer to the answer to Q2 in the general response.
>
>
> **Q5. In the proposed ECQL, there are two main modules. However, there is no experiments studying the importance of these two parts. The authors are suggested to perform an ablation study.**
>
> Thanks for the suggestion. Please refer to the answer to Q3 in the general response for the detailed results.
>
>
> **Q6. The proposed framework integrates a sequential encoder, an actor-critic network, and an evidence network to provide end-to-end integrated training process. For the actor-critic network and evidence network, which one is more important? Moreover, whether the model performance is dominated by the base sequential encoder? Whether the proposed framework can help improve the performance of different base sequential encoders?**
>
> Thanks for the great suggestion. First, we emphasize that the actor-critic network and evidence network work seamlessly to form our ECQL recommender system. Actor-critic provides the RL-base mechanism, while the evidence network provides the evidential prediction and the corresponding vacuity used for effective exploration. Please refer to the answer to Q3 in the general response for the detailed results for impacts of exploration strategies and base sequential encoders.
>
>
>
> **References**
>
> -  Xue, Wanqi, et al. "ResAct: Reinforcing long-term engagement in sequential recommendation with residual actor." ICLR, 2023.
>
> - Antaris, Stefanos, and Dimitrios Rafailidis. "Sequence adaptation via reinforcement learning in recommender systems." Proceedings of the 15th ACM Conference on Recommender Systems. 2021.
>
> - Wang, Xiang, et al. "Neural graph collaborative filtering." Proceedings of the 42nd international ACM SIGIR conference on Research and development in Information Retrieval. 2019.

---

> ### Author Response · Authors · 2023-11-22
>
> Dear Reviewer GVwA,
>
> Thanks for your insightful suggestions to help us further improve the paper! We summarize our update as below:
>
> 1. As suggested, we include two recent RL-based sequential recommendation baselines: : ResAct (Xue et al., 2023) and SAR (Antaris et al., 2021). And we compare them with our approach on two larger datasets, MovieLens-10M and Amazon Book. Please refer to our answers to Q1 and Q2 in the general response for details.
>
> 2. We study the importance of these two modules SSE and CEAC in the proposed framework and conduct ablation studies on both of them. Please refer to the answer to Q3 in the general response for the detailed results.
>
> 3. We have provided the qualitative analysis of diversity in Table 12 of the Appendix. To further provide the quantitative evidence, we calculate the Gini index of the diversity for all the recommended results and conduct an analysis by comparing to other RL-based exploration strategies SAC and CoLin.
>
> We hope you find our answers satisfactory, and consider updating your assessment accordingly! As always, we are happy to provide any additional clarifications if needed.

---

### Official Review · Reviewer_jjM8 · 2023-11-14

**Soundness:** 2 fair
**Presentation:** 3 good
**Contribution:** 2 fair
**Rating:** 5
**Confidence:** 4

**Summary:**

The paper proposes a novel recommendation model that integrates reinforcement learning with evidential learning to provide uncertainty-aware diverse recommendations that may reflect users’ long-term interests. The proposed model, called Evidential Conservative Q-learning (ECQL), conducts evidence-aware explorations to discover items that are located beyond current observation but reflect users’ long-term interests. Additionally, it provides an uncertainty-aware conservative view on policy evaluation to discourage deviating too much from users’ current interests. The paper presents a theoretical analysis to justify the desired convergence behavior and recommendation quality that guarantees to avoid risky (or overly optimistic) recommendations. The paper proposes a framework that includes a sequential state encoder and a Conservative Evidential Actor-Critic (CEAC) module. The former generates the current state of the environment by aggregating historical information and a sliding window that contains the current user interactions as well as newly recommended items from RL exploration that may represent future interests. The latter performs an evidence-based rating prediction by maximizing the conservative evidential Q-value and leverages an uncertainty-aware ranking score to explore the item space for a more diverse and valuable recommendation. The paper conducts experiments over four real-world datasets and compares with state-of-the-art baselines to demonstrate the effectiveness of the proposed model.

**Strengths:**

The paper proposes a novel recommendation model that integrates reinforcement learning with evidential learning to provide uncertainty-aware diverse recommendations that may reflect users’ long-term interests.
    The paper presents a theoretical analysis to justify the desired convergence behavior and recommendation quality that guarantees to avoid risky (or overly optimistic) recommendations.
    The paper conducts extensive experiments over four real-world datasets and compares with state-of-the-art baselines to demonstrate the effectiveness of the proposed model.

**Weaknesses:**

Many of the design choices of the paper are not well justified, it is also quite difficult to understand how some of the components of the model stick together.
    The evaluation of the paper is done on data that is inherently not sequential, (with the exception of the Music data), in particular the Movielens dataset is a survey dataset and has many issues with regards to sequential evaluation.
    Most of the increases in IR scores are rather marginal and it is unclear how much of that can be attributed to the particular elements of the method.
Using ratings as rewards signals is rather difficult to justify as currently ratings are not available for almost all industry related systems

**Questions:**

How does the method compare on real sequential data where reward signals are much more subtle?
What element of the model provides the increase in performance?

---

> ### Author Response · Authors · 2023-11-19
> **Response to Reviewer jjM8 [Part I]**
>
> Thank you for reviewing our paper and providing valuable comments. We summarize our response as follows.
>
> **Q1. Many of the design choices of the paper are not well justified, it is also quite difficult to understand how some of the components of the model stick together.**
>
> Thank you for the comment. The question consists of two parts and we address them separately in what follows.
>
> - *Justification of design choices:* The proposed ECQL seamlessly integrates two major components: a **sequential state encoder** and a **Conservative Evidential Actor-Critic (CEAC)** module. The former primarily focuses on generating the current state of the RL environment by aggregating the previous state, the current items captured by a sliding window, and the future items from the recommendation. This provides an effective means of dynamic state representation for better future recommendations. Meanwhile, the CEAC module leverages evidential uncertainty to effectively explore the item space to recommend items that potentially align with the user's long-term interest. It encourages learning the optimal policy by maximizing a novel conservative evidential Q-value to make more diverse recommendations that may reflect a long-term interest while keeping a conservative view that does not deviate too much from current interests.
>
> - *How the components of the model work together:* As shown in Figure 2 in the paper, the key components of the model include the **sequential state encoder (SSE), Action network, Evidence network**, and **Critic network**. The **SSE** maintains a dynamic state space with a sliding window $W_t$ which moves along the user's interaction history $H_u$ over time to input new data into the SSE, and a conservative evidential-actor-critic (CEAC) module which functions as an RL agent to explore the item space by introducing the evidence-based uncertainty (i.e., vacuity) into a new off-policy evidential RL setting. By incorporating previous state information, recent items captured by a sliding window, and the recommended items from the RL agent, the sequential encoder generates the current state ${\bf s}_t$. This state is further passed to the **Action network** that predicts the mean and variance to form a Gaussian policy distribution. We sample a current action ${\bf a}_t$ from the policy distribution that corresponds to the latent preference of the user that simultaneously captures the past (via a previous state), current (through a sliding window) and future interest (through RL exploration). By leveraging the current action and total item embeddings from the Item Pool ($\mathcal{I}$), the **Evidence network** provides the evidence that can be used to form the rating prediction for exploitation while estimating the uncertainty for better exploration. Finally, the **Critic network** generates a conservative evidential Q-value for conservative policy updates of the action network.
>
> **Q2.  The evaluation of the paper is done on data (except Music) that is inherently not sequential**
>
> We would like to clarify that, broadly speaking, sequential data refers to the data in which current time data is dependent upon its previous time data. For example, the Movielens dataset includes the user interaction sequence in timestamps where the current movie interaction is dependent upon its previous time interactions. In this sense, all the datasets used in our evaluation are sequential as they record how the user interactions with items sequentially changed over a period of time. Furthermore, most of the standard sequential recommendation models like CASER (Tang et al.,2018), SASRec (Kang et al., 2018), and BERT4Rec (Sun et al.,2019) have used these datasets for evaluation. Therefore, we leveraged the chosen datasets to compare with those baselines. In addition, we include two other datasets, including the Amazon book-rating dataset, and conducted additional experiments and comparisons with some recent sequential recommendation methods. Please refer to the answer to Q2 in the general response.

---

> ### Author Response · Authors · 2023-11-19
> **Response to Reviewer jjM8 [Part II]**
>
> **Q3. Most of the increases in IR scores are rather marginal and it is unclear how much of that can be attributed to the particular elements of the method.**
>
> Thank you for this comment. We have provided the standard deviation of the proposed ECQL model along with the most competitive baselines from each category in Table 7 of Appendix E.2. The small standard deviation clearly shows the performance advantage over other methods. We further conduct a significance test to compare ECQL with the second best performing baseline CL4SRec. We obtain a p-value of 0.04, which confirms that the performance advantage of ECQL over CL4SRec is statistically significant. We have also improved our ablation study to investigate the impact of the sequential state encoder and the conservative evidential-actor-critic (CEAC) module. The results are reported in our answer to Q3 in the general response. Furthermore, we also our vacuity-guided exploration, we compare with other commonly exploration strategies. The results can be in our answer to Q3 in the general response. All the new ablation study results are also included in Appendix E.3 of the revised paper.
>
> **Q4. Using ratings as rewards signals is rather difficult to justify as currently ratings are not available for almost all industry related systems**
>
> We follow the standard way of applying RL method in RS, where ratings are leveraged to compute reward as did in recent methods ResAct (Xue et al., 2023) and SAR (Antaris et al., 2021). Our vacuity-based exploration is a general strategy that can be used in other types of user feedback other than ratings. Please refer to our answer to Q5 as well.
>
> **Q5. How does the method compare on real sequential data where reward signals are much more subtle?**
>
> We would like to clarify that our evidential reward definition is a general one that can be extended to other types of user feedback beyond ratings. For example, for click based feedback, the ratings can be changed to a binary signal while the vacuity term (which is the key novelty of our approach) in Equation 3 remains unchanged.
>
> **Q6. What element of the model provides the increase in performance?**
>
> Please refer to the answer to Q3 in the general response for the detailed ablation study results.
>
> **References**
>
> -  Xue, Wanqi, et al. "ResAct: Reinforcing long-term engagement in sequential recommendation with residual actor." ICLR, 2023.
>
> - Antaris, Stefanos, and Dimitrios Rafailidis. "Sequence adaptation via reinforcement learning in recommender systems." Proceedings of the 15th ACM Conference on Recommender Systems. 2021.
>
> - Sun, Fei, et al. "BERT4Rec: Sequential recommendation with bidirectional encoder representations from transformer." Proceedings of the 28th ACM international conference on information and knowledge management. 2019.
>
> - Kang, Wang-Cheng, and Julian McAuley. "Self-attentive sequential recommendation." 2018 IEEE international conference on data mining (ICDM). IEEE, 2018.
>
> - Tang, Jiaxi, and Ke Wang. "Personalized top-n sequential recommendation via convolutional sequence embedding." Proceedings of the eleventh ACM international conference on web search and data mining. 2018.

---

> > ### Author Response · Authors · 2023-11-22
> >
> > Dear Reviewer jjM8,
> >
> > In responding to the raised concerns, we believe the paper has been significantly strengthened, and we thank you for that. In particular, we have clarified
> >
> > 1. Justification of design choices of the **sequential state encoder** and the **Conservative Evidential Actor-Critic (CEAC)** module as well as how these two modules work together seamlessly by maintaining a dynamic state space and conducting effective exploration; conducting an ablation study to investigate the impact of the sequential state encoder and the CEAC module and reporting results in our answer to Q3 in the general response.
> >
> > 2. Explanation of why all our datasets are sequential indeed as they record how the user interactions with items sequentially changed over a period of time.
> >
> > 3. Providing the standard deviation of the proposed ECQL model along with the most competitive baselines from each category to justify the performance advantage is statistically significant and conducting a significance test by comparing the second best baseline "CL4SRec" and report the (small) p-value.
> >
> >
> > We hope you find our answers satisfactory, and consider updating your assessment accordingly! As always, we are happy to provide any additional clarifications if needed.

---

### Author Response · Authors · 2023-11-19
**General response [Part I]**

In this general response, we address a set of important questions that commonly occur in multiple reviewers' comments, avoiding repeating the same response in each individual rebuttal.

**Q1: Lack of comparison with recent works including both sequential recommendation systems and RL based models. (To reviewers GVwA and gFnt)**

We have incorporated the discussion of recent sequential recommendation methods, including: $S^3$-Rec (Zhou et al.,2020) and CL4SRec (Xie et al., 2022), into the related work section of the revised paper. In additional results section of the Appendix from the revised paper, we compare them with ours and show a clear performance gap with statistical guarantees. We also include more recent RL models, including ResAct (Xue et al., 2023) and SAR (Antaris et al., 2021), which are also developed for sequential recommendations.  The comparison of these recent work with our proposed model is conducted on two larger datasets. Please refer to our response to Q2 below for details.

**Q2: Datasets used for evaluation are small. (To reviewers GVwA and gFnt)**

As suggested, we incorporate Movielens-10M, which has around 70K users and 11K items with 10M interactions. Furthermore, we have added the Amazon Book rating dataset (Wang et al., 2019), which has more than 52K users and 91K items with 3M ratings. We include the comparison with two newer RL-based sequential recommendation baselines and the results are reported in the table below. As can be seen, ECQL consistently outperforms these newer baselines and achieves solid recommendation performance on larger datasets.

| **Model**     | **ML-10M**| |**Amazon Book**||
|  -- | -----------  |--| -----------  |--|
|     | **P@5** |**nDCG@5** |**P@5** |**nDCG@5** |
| SAR | 0.5374|0.4762 |0.4915 | 0.3327 |
| ResAct |0.5624 |0.5136 |0.5273 |0.3654 |
|**ECQL**|**0.6425$\pm$0.025**|**0.5518$\pm$0.022**|**0.6017$\pm$0.024**|**0.4896$\pm$0.018**|

 For more details about the datasets, please refer to Appendix E.1 and for result analysis, please refer to the Appendix E.2 of the revised paper.

---

> ### Author Response · Authors · 2023-11-19
> **General response [Part II]**
>
> **Q3: Additional ablation study to show (1) the contribution of each module, (2) vacuity-based exploration strategy, and (3) dependency of the sequential encoder. (To reviewers GVwA and jjM8)**
>
> - **(1) Ablation study on each module: the sequential state encoder and the conservative evidential actor-critic (CEAC) module.** We evaluate the impact of each module by training the model with "RNN", "CEAC" and "RNN+CEAC" configurations as detailed below:
>
>     - "RNN" means we use RNN as our sequential encoder and use an MLP based classifier to predict the rating for each item in the item pool for both training and test user sets.
>
>     - "CEAC" means that we only leverage current user embedding to be the action network's input and continue the RL process without any sequential state encoder to help maintain the state embedding.
>
>     - "RNN+CEAC" means that two modules work together as our ECQL framework.
>
>     We calculate the P@5 and nDCG@5 for each setting in the Movielens-1M test set. The results as summarized in the table below (also in Appendix E.3 of the revised paper) show that the integration of the two modules significantly outperform each individual module.
>
>     | **Modules**     | **P@5**|**nDCG@5**|
>     |  -- | -----------  |--|
>     | RNN | 0.4899 | 0.4175|
>     | CEAC |0.6044 | 0.5108 |
>     |**RNN+CEAC**|**0.6313**|**0.5365**|
>
> - **(2) Ablation study on vacuity-based exploration strategy**. To further demonstrate the important role of our novel evidential design, we compare different model designs with varying exploration strategies:
>
>     - $\epsilon$-greedy (AC, QL) is a basic exploration strategy, which uses actor-critic to conduct Q-learning. Since we only set hyper-parameter $\epsilon=0.1$, which is our best practice after grid-search, there's no need to combine conservative learning in this exploration direction.
>
>     - SAC also leverages actor-critic to conduct Q-learning. The difference comparing to $\epsilon$-greedy is that it uses first-order uncertainty "Entropy" in the rating prediction for score-based exploration and the reward design.
>
>     -  EQL is our framework with traditional Q-learning in stead of conservative Q-learning, so the exploration is purely guided by uncertainty without any constraints.
>
>     - CoLin is a collaborative contextual bandit based method which explicitly models the underlying dependency among users/bandits with the help of a weighted adjacency graph, comparing to other bandit exploration technique like upper confidence bound based (UCB-based) HLin or factorUCB, it achieves best performance in multiple recommendation datasets.
>
>     We report the comparison performance in Movielens-1M test set in the table below (also in Appendix E.3 of the revised paper). The results demonstrate a clear advantage of the proposed exploration strategy over other baselines.
>
>     | **Models**     | **P@5**|**nDCG@5**|
>     |  -- | -----------  |--|
>     $\epsilon$-greedy (AC, QL)| 0.5977 | 0.4834|
>     SAC| 0.6105 | 0.5215|
>     EQL| 0.6234 | 0.5331|
>     CoLin | 0.6162 | 0.5216 |
>     ECQL |0.6313 | 0.5365 |
>
> - **(3) Ablation study on the sequential encoder**.  We have performed an ablation study on three base sequential encoders (RNN, LSTM, and GRU) along with the proposed CEAC module on the MovieLens-1M test set to analyze their impact on the training process.
>
>     | **Models**     | **P@5**|**nDCG@5**|
>     |  -- | -----------  |--|
>     | RNN+CEAC | 0.6313 | 0.5365|
>     | LSTM+CEAC |0.6312 | 0.5365 |
>     |GRU+CEAC|0.6315|0.5366|
>
>     The result (also in Appendix E.3 of the revised paper) shows that our model's performance remains robust to the type of the sequence encoder.

---

### Meta-Review · Area_Chair_v76i · 2023-12-14

**Metareview:**

The paper proposes a new recommendation model that integrates a sequential state encoder and a Conservative Evidential Actor-Critic (CEAC) module to provide uncertainty-aware diverse recommendations that may reflect users’ long-term interests. The proposed model, called Evidential Conservative Q-learning (ECQL), conducts evidence-aware explorations to discover items that are located beyond current observation but reflect users’ long-term interests. Additionally, it provides an uncertainty-aware conservative view on policy evaluation to discourage deviating too much from users’ current interests.

Strength: the paper presents a theoretical analysis to justify the desired convergence behavior and recommendation quality that guarantees to avoid risky (or overly optimistic) recommendations.
Weakness: multiple reviewers raised questions on the scale and sequential aspects of the chosen datasets and experimental evaluations. Reviewers asked for ablation studies to understand the effect of different components and different design choices made, and comparison to more recent work in this area.

**Justification For Why Not Higher Score:**

While reviewers appreciate the overall framework, many asked for ablation studies and insights to understand the effect of different components (i.e., the sequential encoder, the actor-critic network, and the evidence network and uncertainty-aware policy evaluation) and different design choices made.  Multiple reviewers raised questions on the scale and sequential aspects of the chosen datasets and experimental evaluations, and asked for comparison to more recent works. While the authors did update the experiments during rebuttal, reviewers were not able to vet them. The large number of added experiments and ablations will call for another round of reviews.

**Justification For Why Not Lower Score:**

N/A

---

### Decision · Program_Chairs · 2024-01-16

Reject